# LABO: LLM-Accelerated Bayesian Optimization through Broad Exploration and Selective Experimentation

**Zhuo Chen** [* 1 2]  **Xinzhe Yuan** [* 3 1]  **Jianshu Zhang** [1 4]  **Jinzong Dong** [1 5]  **Ruichen Zhou** [6]  **Yingchun Niu** [6]
**Tianhang Zhou** [7]  **Yu Yang Fredrik Liu** [8]  **Yuqiang Li** [1]  **Nanyang Ye** [1 4]  **Qinying Gu** [1]

## Abstract

The high cost and data scarcity in scientific exploration have motivated the use of large language models (LLMs) as knowledge-driven components in Bayesian optimization (BO). However, existing approaches typically embed LLMs directly into the sampling or surrogate modeling pipeline, without fully leveraging their significantly lower evaluation cost compared to real-world experiments. To address this limitation, we propose LLM-Accelerated Bayesian Optimization (LABO), a framework that combines LLM predictions with experimental observations within a single BO loop. LABO employs a gating criterion to dynamically balance the reliance on LLM predictions versus actual experiments. By leveraging inexpensive LLM evaluations to broadly explore the search space and reserving costly real experiments only for regions with high uncertainty, LABO achieves more sample-efficient optimization. We provide a theoretical analysis with a cumulative regret bound that formalizes this efficiency gain. Empirical results across diverse scientific tasks demonstrate that LABO consistently outperforms existing methods under identical experimental budgets. Our results suggest that LABO offers a practical and theoretically grounded approach for integrating LLMs into scientific discovery workflows.

## 1. Introduction

Formulation optimization is a central, yet challenging task in scientific domains such as drug discovery (Zhang et al., 2025a), catalyst design (Xin, 2022), and molecular engineering (Chen et al., 2025). Unlike conventional optimization tasks in machine learning, each evaluation in formulation design typically requires costly, time-consuming, and labor-intensive experiments (Wang et al., 2023). These constraints make exhaustive exploration impractical under limited budgets. Moreover, the scarcity of experimental data limits the applicability of data-hungry methods. To address these challenges, Bayesian optimization (BO) has become a key strategy for navigating complex design spaces (Shahriari et al., 2015; Shields et al., 2021; Guo et al., 2023; Yuan et al., 2024). By modeling the objective function with a surrogate and selecting candidates using acquisition functions, BO balances exploration and exploitation to efficiently identify promising solutions (Shields et al., 2021).

Despite its widespread success, BO continues to face two fundamental limitations. First, data scarcity at the beginning of optimization impedes early-stage exploration, a challenge commonly known as the cold-start problem (Guo et al., 2023). Second, exploration in high-dimensional design spaces remains challenging for BO under limited experimental budgets (Shahriari et al., 2015; Shields et al., 2021). Although properly configured vanilla BO can remain competitive in some high-dimensional settings (Hvarfner et al., 2024a), expensive scientific formulation tasks still pose practical challenges due to sparse task-specific observations and limited real-fidelity budgets. These issues together constrain the sample efficiency of BO and limit its effectiveness in scientific discovery. Notably, most scientific problems are rich in domain knowledge, often embedded in textual resources such as literature, protocols, and expert guidelines. Human experts often leverage this information to guide the selection of sampling points (Hvarfner et al., 2024b). Inspired by this, effectively integrating such knowledge as prior knowledge into the BO framework offers a promising pathway toward

---

[*]Equal contribution  [1]Shanghai Artificial Intelligence Laboratory, Shanghai, China [2]School of Mechanical Engineering, Shanghai Jiao Tong University, Shanghai, China [3]Institute for Advanced Study in Mathematics, Harbin Institute of Technology, Harbin, China [4]School of Computer Science, Shanghai Jiao Tong University, Shanghai, China [5]School of Automation, Central South University, Changsha, China [6]College of New Energy and Materials, China University of Petroleum, Beijing, China [7]College of Carbon Neutrality Future Technology, China University of Petroleum, Beijing, China [8]DeepVerse PTE. LTD., Singapore. Correspondence to: Qinying Gu <guqinying@pjlab.org.cn>, Nanyang Ye <ynylincoln@sjtu.edu.cn>, Tianhang Zhou <zhouth@cup.edu.cn>.

*Proceedings of the 43rd International Conference on Machine Learning*, Seoul, South Korea. PMLR 306, 2026. Copyright 2026 by the author(s).

more efficient and informed optimization.

Indeed, recent efforts have sought to bridge this gap by integrating large language models (LLMs) into the BO pipeline to incorporate prior knowledge. Existing methods typically use LLMs to provide guidance at the initialization stage (Liu et al., 2024b; Agarwal et al., 2025) or to offer additional suggestions (Yin et al., 2024; Yang et al., 2025b; Chang et al., 2025) during the optimization process. In these approaches, LLMs are primarily queried to evaluate individual candidates or to refine local decisions based on prior knowledge and accumulated observations. However, LLMs are capable of producing predictive assessments through in-context reasoning over prior knowledge and historical observations (Yang et al., 2023). Crucially, such predictions can be obtained at a cost that is orders of magnitude lower than real-world experiments. Yet current approaches make limited use of this cost-effective information, which restricts their ability to support broad, low-cost exploration across the search space. These insights motivate a central question:

> *Can LLMs be systematically incorporated into the optimization process to enable more effective exploration and improve the efficiency of BO?*

In this paper, we answer this question affirmatively by proposing LLM-Accelerated Bayesian Optimization (LABO). LABO instantiates a dual-fidelity BO workflow that integrates LLM-fidelity predictions with real-fidelity observations. In the warm-start phase, LABO leverages the LLM to propose high-potential candidates for early real-fidelity measurements, and to generate a broad set of LLM-fidelity predictions that cover the search space at negligible cost. During optimization, LABO fits a Kennedy–O'Hagan (KOH) joint Gaussian process (GP) surrogate (Kennedy & O'Hagan, 2000), decomposing the real-fidelity objective into a scaled LLM-fidelity component and a discrepancy process. LABO then applies a discrepancy-dominance gating criterion to determine whether a candidate should trigger a real-fidelity experiment, based on how much predictive uncertainty is driven by the discrepancy term.

We support this design with both theory and experiments. Theoretically, we provide regret guarantees showing that LABO is sample-efficient when LLM guidance is informative and remains robust under noisy or misaligned LLM-fidelity signals (under the KOH modeling assumptions). Empirically, across a range of scientific optimization tasks, LABO finds better candidates under the same real-fidelity budget when LLM predictions are helpful, and matches vanilla BO when they are not.

Our contributions are threefold:

1. We propose **LABO**, a principled BO framework that integrates LLM-derived predictions alongside real experiments.

2. We establish theoretical guarantees showing that LABO improves sample efficiency when LLM guidance is informative, and bounds the regret even when the LLM-fidelity signal is misleading.

3. We demonstrate the practical utility of LABO across diverse scientific optimization tasks, where it discovers high-performing formulations more efficiently under fixed real-fidelity experimental budgets.

**Conflict of Interest Disclosure.** The authors Z. C., X.Y., J.Z., J.D., N.Y., and Q.G. are employed by Shanghai Artificial Intelligence Laboratory, which leads the development of Intern S1 and Intern-S1-mini, which were among the LLMs evaluated in this paper.

## 2. Related Work

Currently, BO approaches that involve LLMs can be broadly grouped into two lines of thought:

**LLM-in-the-loop BO.** This family of methods leverages LLMs to improve components of the BO pipeline, such as the search space, the surrogate model, or the sampling/acquisition strategy (Liu et al., 2024b; Agarwal et al., 2025; Yang et al., 2025b; Suwandi et al., 2025; de Carvalho et al., 2025; Yuan et al., 2026). LLAMBO (Liu et al., 2024b) leverages LLMs to generate high-potential initialization points and propose candidate solutions during optimization, alleviating cold-start issues. However, final decisions remain governed by conventional acquisition functions, with the LLM only indirectly involved in the optimization loop. Subsequent studies follow similar paradigms. BOPRO (Agarwal et al., 2025) performs BO in a latent space induced by LLM embeddings, decoding optimized latent targets into executable solutions via prompting. ReasoningBO (Yang et al., 2025b) improves proposal quality by incorporating chain-of-thought prompting. CAKE (Suwandi et al., 2025) takes a different approach by injecting LLM-derived priors into the kernel of a GP, allowing semantic knowledge to shape the surrogate model. FIBO (de Carvalho et al., 2025) leverages a pretrained generative model to accelerate acquisition optimization and sampling within BO, reducing computational overhead per iteration.

**LLM as an external signal.** These methods use LLMs to estimate or predict the performance of candidate inputs, and then incorporate such predictions into the BO loop (Zhang et al., 2025b; Ballew et al., 2025; Han et al., 2025; Menet et al., 2025). ChemBOMAS (Han et al., 2025) further uses LLMs to run pseudo-experiments and injects the predicted outcomes as initial observations, accelerating early-stage

exploration in chemical optimization. ToSFiT (Menet et al., 2025) reformulates Thompson sampling in large discrete spaces as an online LLM fine-tuning process, using fine-tuned LLMs to guide exploration over discrete candidates.

However, while informative, prior methods either assign LLMs only partial responsibility within the loop or restrict them to a signal source mainly at initialization. This design overlooks the LLM's capacity for broad and diversified exploration.

## 3. Preliminary

Before introducing our approach, we briefly review the standard formulation of BO, along with a commonly used modeling assumption for combining heterogeneous information sources. In particular, the KOH assumption provides a formulation for relating one signal to another.

**Vanilla BO.** BO aims to maximize an unknown function $f : \mathcal{X} \to \mathbb{R}$ by sequentially querying inputs $x \in \mathcal{X}$, where evaluations are expensive or limited. At iteration $t$, a set of observations $D_t = \{(x_i, y_i)\}_{i=1}^{n_t}$ is used to fit a probabilistic surrogate model, commonly a GP with mean function $\mu$ and kernel $k$. This surrogate yields a predictive distribution over $f(x)$, including both a mean estimate and associated uncertainty. An acquisition function $a_t(x)$, such as expected improvement (Jones et al., 1998) or upper confidence bound (UCB) (Srinivas et al., 2009), is computed from the surrogate and optimized to select the next query point. By updating the surrogate after each observation, BO efficiently balances exploration and exploitation to locate the global optimum with minimal evaluations.

**KOH assumption (Kennedy & O'Hagan, 2000).** The KOH assumption is a foundational modeling principle in multi-fidelity emulation. It posits that a real-fidelity output $f_R(\mathbf{x})$ can be decomposed into a linear transformation of an LLM-fidelity approximation $f_L(\mathbf{x})$ and an additive discrepancy term $\delta(\mathbf{x})$ , i.e.,

$$f_R(x) = \rho f_L(x) + \delta(x), \qquad (1)$$

where $\rho$ is a scalar representing the calibration or scaling factor between fidelities, and $\delta(\mathbf{x})$ captures structural differences or biases not accounted for by the LLM-fidelity model. This formulation enables the integration of information across fidelity levels while explicitly accounting for model inadequacy.

## 4. Method

In scientific optimization, human experts often select promising experimental parameters based on domain knowledge and past experience. Such expert-driven suggestions,

though not derived from physical measurements, can effectively guide the search process and, in essence, constitute a distinct source of fidelity information (Brochu et al., 2010; Xu et al., 2024). Inspired by this, we treat modern LLMs as a practical interface to unstructured scientific priors, including literature, protocols, and engineering heuristics. By combining these priors with the accumulated optimization history via in-context reasoning, an LLM can produce fast predictions of candidate designs. Importantly, these predictions are valuable not merely because they are inexpensive to obtain, but also because they can encode broad and cross-disciplinary scientific knowledge that provides useful intuition and directional guidance. This distinguishes LLM-fidelity information from conventional simulation-based models, which often require task-specific model construction, domain expertise, and substantial development effort for each new problem.

Building on this perspective, we propose LLM-Accelerated Bayesian Optimization, a framework designed to improve the sample efficiency of BO by systematically integrating LLM-derived predictions with real-fidelity experimental measurements. This dual-fidelity setting introduces two core challenges: (i) how to fuse heterogeneous LLM- and real-fidelity signals into a unified probabilistic surrogate, and (ii) how to decide, at each step, whether a candidate should be evaluated using LLM predictions or through an expensive real-fidelity experiment.

LABO addresses these challenges with two key strategies: On the modeling side, we employ discrepancy modeling based on the KOH assumption to correct systematic biases between LLM- and real-fidelity evaluations. On the decision side, we introduce a gating criterion to control the influence of LLM predictions, using uncertainty-based gating to dynamically determine when to rely on LLM estimates or real-fidelity evaluations. These strategies enable LABO to leverage LLM guidance while preserving robustness and ensuring efficient use of the real-fidelity budget. Section 4.1 introduces the LLM-aware modeling approach, followed by the gating criterion in Section 4.2. The overall optimization procedure is provided in Section 4.3.

### 4.1. LLM-Aware Modeling

LABO treats LLMs as *knowledge-driven* information sources. To integrate such cognitive priors with real experimental data, we adopt the KOH joint GP framework. Specifically, we model $f_L(x)$ and $\delta(x)$ as independent zero-mean GPs:

- $f_L(x) \sim \mathcal{GP}(0, k_L(x, x'))$ , trained on LLM predictions;

- $\delta(x) \sim \mathcal{GP}(0, k_\delta(x, x'))$, trained on the discrepancy between real-fidelity observations and $\rho f_L(x)$.

Here, $k_L$ and $k_\delta$ are positive definite kernel functions. The scaling factor $\rho$ is estimated by ordinary least squares over paired LLM- and real-fidelity observations, providing a simple calibration of the global relationship between the two fidelities. With $\rho$ fixed, the resulting KOH covariance is well defined for any positive definite $k_L$ and $k_\delta$, while the discrepancy process $\delta(x)$ captures the remaining input-dependent bias that cannot be explained by the scaled LLM-fidelity function. This formulation yields a posterior over the real-fidelity function, with predictive mean and variance at any input $x$ given by:

$$\mu_R(x) = \rho\mu_L(x) + \mu_\delta(x), \quad \sigma_R^2(x) = \rho^2\sigma_L^2(x) + \sigma_\delta^2(x).$$

This modeling structure treats LLM predictions as a base estimator and uses the discrepancy process to correct for non-physical biases inherent in language-model-based reasoning. Crucially, because the LLM-aware model couples $f_L$ and $f_R$, incorporating additional LLM queries (even without new experiments) updates $(\mu_L, \sigma_L^2)$ and thereby refines the real-fidelity posterior $(\mu_R, \sigma_R^2)$. Intuitively, when LLM predictions align well with observed outcomes, the discrepancy remains small, allowing the surrogate to leverage LLM priors efficiently; when predictions become unreliable, the discrepancy variance dominates the total uncertainty, automatically triggering the need for real-world evaluation. This enables LABO to safely extract value from a *cognitive*, knowledge-based LLM-fidelity source, rather than relying on traditional numerical simulation surrogates.

### 4.2. Core Tool: Gating Criterion

LABO uses a gating criterion to determine whether an LLM-fidelity prediction is sufficiently reliable to update the surrogate without requiring a real-fidelity query. To quantify the reliability of the LLM-fidelity prediction, we define the *discrepancy dominance ratio*:

$$p_\Delta(x) = \frac{\sigma_\delta^2(x)}{\rho^2\sigma_L^2(x) + \sigma_\delta^2(x)},$$

which measures the proportion of total predictive uncertainty in the real-fidelity surrogate attributable to the discrepancy process $\delta(x)$. A larger value of $p_\Delta(x)$ indicates that the uncertainty associated with the discrepancy dominates, suggesting that the relationship between LLM- and real-fidelity observations is difficult to model at $x$, and that the real-fidelity response cannot be reliably inferred from the LLM-fidelity prediction.

We adopt a threshold-based decision rule with a gating parameter $\tau \in (0, 1)$. If $p_\Delta(x) \leq \tau$, the LLM-fidelity prediction is deemed reliable and used to update the joint surrogate via the LLM-fidelity GP of $f_L(x)$. Otherwise, LABO performs a real-fidelity experiment and updates the GPs of both $f_L(x)$ and $\delta(x)$. Equivalently, LABO triggers an experiment only when the surrogate uncertainty is dominated

by the discrepancy, i.e., when LLM-fidelity information is insufficient to reduce uncertainty at that location. This rule allows LABO to exploit LLM evaluations where they are useful, while reserving experiments for regions where LLM-fidelity information is insufficient. As a result, LABO significantly reduces the number of real-fidelity queries while maintaining accurate surrogate modeling.

### 4.3. Optimization Workflow

The overall optimization workflow of LABO is illustrated in Figure 1 and summarized in Algorithm 1. The workflow begins with a warm-start phase to initialize the optimization process, followed by an iterative loop that selectively incorporates real evaluations based on the informativeness of LLM predictions.

**Prior-guided initialization.** LABO initializes the optimization by engaging the LLM to reason based on task-specific prior knowledge, including domain insights from literature, physical feasibility constraints, and semantic descriptions of target properties. This knowledge-driven approach serves two distinct purposes: to mitigate cold-start limitations and to establish a broad, informed view of the search space before the optimization loop begins. To address the first goal, the LLM synthesizes its priors to propose a small set of candidates $\mathcal{X}_R$ with high exploratory value under the given constraints. These inputs are evaluated through real-fidelity experiments to obtain ground-truth measurements $y_R(x)$, forming the initial dataset $\mathcal{D}_R = \{(x, y_R(x)) \mid x \in \mathcal{X}_R\}$. In parallel, a batch of space-covering inputs $\mathcal{X}_L$ is selected using Latin Hypercube Sampling (LHS). The LLM is queried at each $x \in \mathcal{X}_L$ to produce low-cost predictions $y_L(x)$, forming the initial LLM-fidelity dataset $\mathcal{D}_L = \{(x, y_L(x)) \mid x \in \mathcal{X}_L\}$. These predictions support the construction of $f_L$ and provide coarse-grained guidance on the global structure of the objective landscape. To enable discrepancy modeling in later stages, LLM predictions are also obtained at all $x \in \mathcal{X}_R$, ensuring that $\mathcal{X}_R \subset \mathcal{X}_L$. The complete input information provided to the LLM during this phase is included in Appendix.

**Optimization loop.** Following initialization, LABO advances the optimization through a dynamic loop that prioritizes the use of LLM-fidelity predictions while invoking real-fidelity evaluations only when necessary. At an iteration $t$, LABO first constructs a real-fidelity surrogate $f_R(x)$ that integrates low-cost predictions with experimentally derived corrections. The LLM-fidelity surrogate $f_L(x)$ is trained on the dataset $\mathcal{D}_L$ and provides a coarse but structured approximation of the objective function. To account for systematic bias, LABO estimates a scaling factor $\rho$ by minimizing the squared error between $y_R(x)$ and $y_L(x)$ over all inputs with

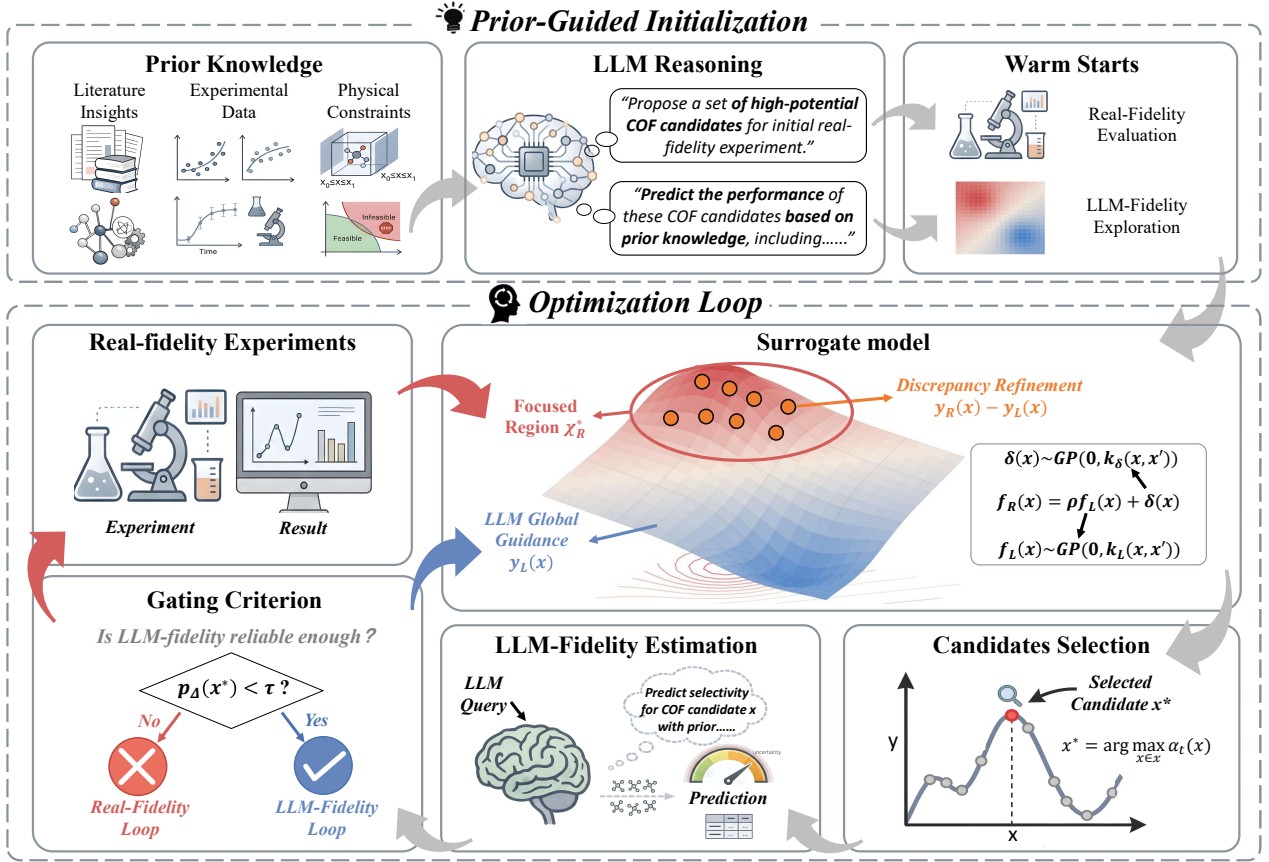

Figure 1. Overview of the LABO workflow. Initialization combines prior knowledge and LLM reasoning to propose high-potential points and perform LLM-fidelity exploration. Each iteration updates a joint surrogate, selects candidates, queries the LLM-fidelity predictions, and applies a gating criterion to decide whether to query the real-fidelity source. The process flow is illustrated with different color codes: **grey arrows** represent the general workflow, **blue** ones indicate the process when LLM-fidelity predictions are reliable, and **red** ones denote the process when LLM-fidelity predictions are unreliable, triggering the need for real-fidelity queries.

paired LLM- and real-fidelity observations. It then constructs a discrepancy model $\delta(x)$ using the residual dataset $\{(x, y_R(x) - \rho y_L(x)) \mid (x, y_R(x)) \in \mathcal{D}_R\}$. The final surrogate is defined as $f_R(x) = \rho f_L(x) + \delta(x)$ and serves as the basis for candidate selection in the next step.

Based on the surrogate $f_R(x)$, LABO selects a batch of candidate points $\mathcal{X}_t$ by maximizing an acquisition function. To improve batch diversity, we construct the batch in a sequential greedy manner, where the surrogate is temporarily updated with the posterior mean after each selected candidate before selecting the next one. For each $x \in \mathcal{X}_t$, LABO queries the LLM to obtain a low-cost prediction $y_L(x)$, and adds $(x, y_L(x))$ to $\mathcal{D}_L$. The LLM query includes the task description, parameter definitions, optimization objective, historical observations, and the candidate points to be predicted, and the output is constrained to a structured JSON format with task-specific valid target ranges. LABO then evaluates the gating criterion $p_\Delta(x)$ to determine whether a

real-fidelity evaluation is necessary. For candidates that fail the gating criterion, LABO performs a real experiment to obtain $y_R(x)$ and adds the result to $\mathcal{D}_R$.

Once all candidates $\mathcal{X}_t$ are processed, LABO updates its models. The LLM-fidelity surrogate $f_L$ is retrained on the updated $\mathcal{D}_L$, the discrepancy model $\delta(x)$ is updated using the new residuals, and the scaling factor $\rho$ is re-estimated. The composite surrogate $f_R(x)$ is then recomputed accordingly by equation 1. The loop continues until the real-fidelity budget is exhausted or a stopping condition is met.

## 5. Theoretical Guarantee

We establish a theoretical guarantee (Section A) for LABO by analyzing its cumulative regret in the KOH setting. To decide which fidelity to query at each step, LABO employs a gating criterion based on the ratio between the discrepancy variance and the total predictive variance of the real-fidelity

---

**Algorithm 1** LLM-Accelerated Bayesian Optimization

---

1: **Input:** real-fidelity budget $T$, gating threshold $\tau$, prior knowlegde $\mathcal{P}$, LLM evaluator $g_{LLM}(\mathcal{P}, \mathcal{D})$ real world experiment $g_R(\mathcal{X}_R)$ LLM-aware kernel $k_L$ and discrepancy kernel $k_\delta$.
2: **Output:** best $(x^*, y_R^*)$ observed in $\mathcal{D}_R$
3: Query initialization experiments set $\mathcal{X}_R \leftarrow g_{LLM}(\mathcal{P})$
4: Real-world experiments $y_R(x) \leftarrow g_R(\mathcal{X}_R)$
5: Generate LHS samples $\mathcal{X}_{LHS}$ and set $\mathcal{X}_L \leftarrow \mathcal{X}_R \cup \mathcal{X}_{LHS}$, $\mathcal{D}_R = \{(x, y_R(x)) \mid x \in \mathcal{X}_R\}$
6: Query LLM to obtain $\mathcal{D}_L \leftarrow g_{LLM}(\mathcal{P}, \mathcal{D}_R)$
7: **while** the number of real-fidelity evaluations $< T$ **do**
8:     Model $f_L \sim \mathcal{GP}(0, k_L)$ on $\mathcal{D}_L$
9:     Compute $\rho \leftarrow \arg\min_\rho \sum_{(x,y_R(x))\in\mathcal{D}_R}(y_R(x) - \rho y_L(x))^2$
10:     Model $\delta \sim \mathcal{GP}(0, k_\delta)$ on $\mathcal{D}_\delta = \{(x, y_R(x) - \rho y_L(x)) \mid (x, y_R(x)) \in \mathcal{D}_R\}$
11:     Compute $f_R(x) \leftarrow \rho f_L(x) + \delta(x)$
12:     Compute $\mathcal{X}_t \leftarrow \arg\max_x UCB(x, f_R(x))$
13:     **for** each $x \in \mathcal{X}_t$ **do**
14:         Query LLM $\mathcal{D}_L \leftarrow \mathcal{D}_L \cup \{(x, y_L(x))\}$
15:         **if** $p_\Delta(x) > \tau$ **then**
16:             Real-fidelity evaluation to update $\mathcal{D}_R \leftarrow \mathcal{D}_R \cup \{(x, y_R(x))\}$
17:         **end if**
18:     **end for**
19: **end while**

---

model. This criterion identifies whether the LLM-fidelity estimate is sufficiently informative, and adaptively partitions the full design domain $\mathcal{X}$ into regions where LLM-fidelity queries are effective and regions requiring real-fidelity evaluation. We prove that this partition stabilizes after a finite number of steps, giving rise to a limiting real-fidelity region $\mathcal{X}_R^* \subseteq \mathcal{X}$ where real-fidelity queries are consistently selected. Formally, as defined in Definition A.6, the limiting real-fidelity region is

$$\mathcal{X}_R^* := \liminf_{t\to\infty} \mathcal{X}_R^{(t)}.$$

The cumulative regret $R_T = \sum_{t=1}^{T}(f^* - f_R(x_t))$ is then decomposed into three components: regret from real-fidelity queries within $\mathcal{X}_R^*$, regret from a sublinear number of real-fidelity queries outside this region during the transient phase, and regret from LLM-fidelity evaluations. Each term is bounded using standard GP-UCB arguments and the mutual information associated with the induced real-fidelity model. The resulting bound provides an asymptotic cumulative-regret guarantee, implying sublinear regret under the stated assumptions and standard kernel-specific information-gain rates.

Importantly, our analysis places no structural assumptions

on the LLM-fidelity oracle, allowing it to be inaccurate or inconsistent across the domain. This generality makes the result applicable to cases where the external source is an LLM with spatially varying uncertainty. By concentrating real-fidelity queries within a progressively restricted subset of the domain, LABO achieves improved sample efficiency and reduced regret. The proof of Theorem 5.1 is shown in Appendix A.

**Theorem 5.1.** *Suppose $f_R \sim \mathcal{GP}(0, k_R)$ with the composite kernel defined above. Let $T_R^*$ be the number of real-fidelity queries within the limiting region $\mathcal{X}_R^*$, and let $T_L$ be the number of LLM-fidelity queries. Let $\beta_T = \mathcal{O}(\log T)$ be the standard GP-UCB confidence parameter, and let $\Psi_T(A)$ denote the maximum mutual information between $f_R$ and any set of $T$ noisy observations in region $A$. Then, with probability at least $1 - \delta$, the cumulative regret of LABO satisfies*

$$R_T \leq C_1\sqrt{T_R^*\beta_T\Psi_T(\mathcal{X}_R^*)} + C_2\sqrt{T^\alpha\beta_T\Psi_T(\mathcal{X})} + C_3\sqrt{T_L\beta_T\Psi_T(\mathcal{X})},$$

*where $C_1, C_2, C_3 > 0$ are constants, and $\alpha < 1$ reflects the sublinear number of transient real-fidelity queries outside $\mathcal{X}_R^*$.*

**Remark 5.2.** Standard GP-UCB explores the entire domain using a single real-fidelity model, leading to regret that scales with the information capacity $\Psi_T(\mathcal{X})$ of the full search space. In contrast, the gating criterion in LABO rapidly filters out regions where LLM-fidelity signals are uninformative and where real-fidelity evaluations offer no meaningful uncertainty reduction. This yields a substantially reduced effective region $\mathcal{X}_R^*$. Consequently, the dominant regret term is governed by $\Psi_T(\mathcal{X}_R^*) \ll \Psi_T(\mathcal{X})$, yielding significantly more focused exploration and improved sample efficiency. Furthermore, since LLM-fidelity queries often incur negligible cost compared to real scientific experiments in practice, their contribution to the overall regret, $C_3\sqrt{T_L\beta_T\Psi_T(\mathcal{X})}$, can be ignored.

## 6. Experiments

We evaluate the performance of LABO on a suite of continuous formula optimization tasks drawn from multiple scientific domains and of varying dimensionalities. These tasks are designed to support a systematic comparison with vanilla BO and recent LLM-guided baselines under a fixed real-fidelity evaluation budget. In addition to the final best objective value, we also consider how quickly each method reaches high-quality solutions, since each real-fidelity evaluation may correspond to a costly scientific experiment. Faster convergence therefore directly translates into fewer experimental trials and greater practical utility in scientific discovery. Appendix C provides additional evaluations on AutoML benchmarks (Eggensperger et al., 2021),

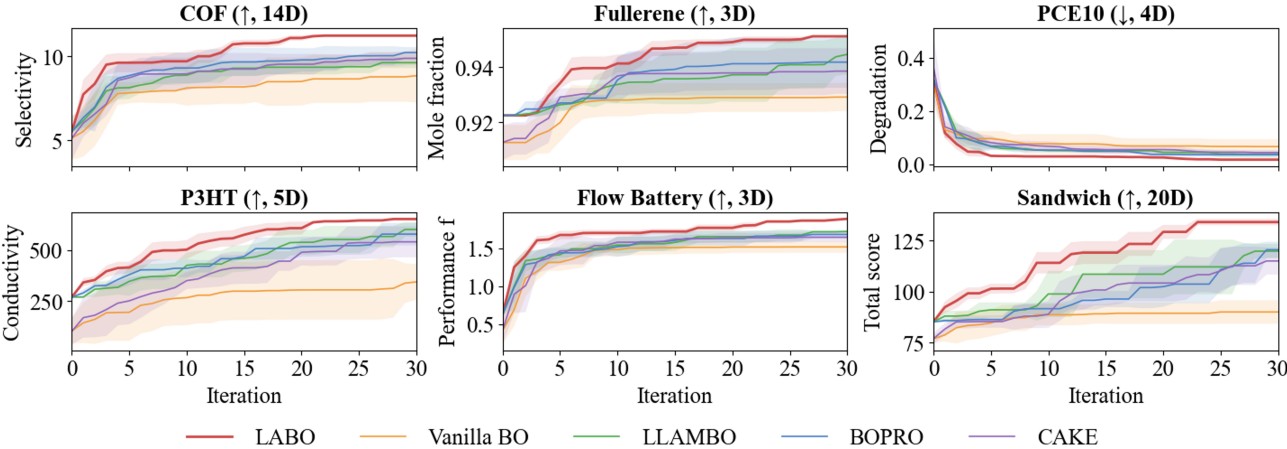

*Figure 2.* Optimization performance across six scientific tasks. Shaded regions indicate standard deviation over five independent seeds.

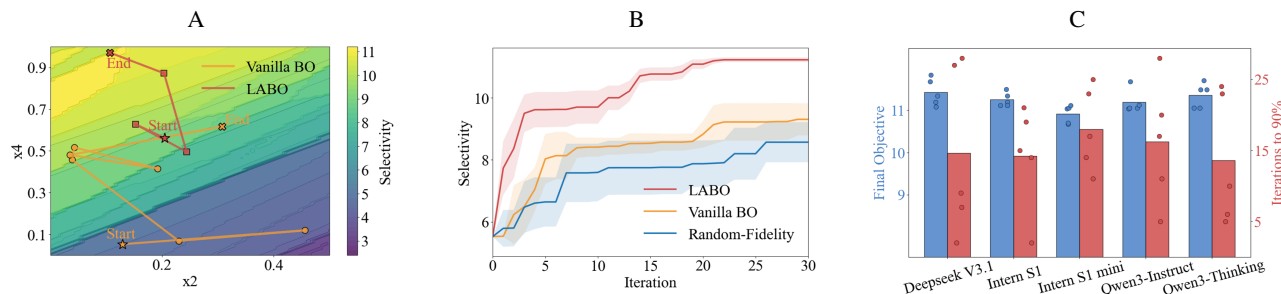

*Figure 3.* Ablation results on the COF task. **(A)** Optimization trajectories of LABO and Vanilla BO. **(B)** Performance comparison under the same initialization candidates across LABO, Vanilla BO, and random-fidelity prediction. **(C)** Performance of LABO with different LLMs. The blue bars represent the final best objective value, while the red bars show the number of iterations required to reach 90% of the final optimum. Dots indicate results from individual test runs.

high-dimensional scientific optimization (Hamidieh, 2018), cost-benefit analysis, and LLM–real-fidelity alignment.

### 6.1. Experimental settings

**Baselines.** We consider the following methods:

**Vanilla BO** uses a GP surrogate and a standard q-UCB acquisition function, without any external knowledge.

**LLAMBO** (Liu et al., 2024b) integrates LLMs by prompting them to suggest initial candidates and generate new proposals during optimization.

**BOPRO** (Agarwal et al., 2025) iteratively updates the LLM prompt using selected histories to sample new candidates that explore or exploit different regions of the search space.

**CAKE** (Suwandi et al., 2025) uses LLMs to adaptively generate and refine GP kernels based on observed data during the optimization process.

**Implementation details.** All methods use a GP surrogate with an RBF kernel and the q-UCB acquisition function. Each iteration selects two candidates for evaluation. Op-

timization begins with three initial points per task. For LABO, a batch of 50 LLM-fidelity evaluations is performed during the warm-up phase before real-fidelity optimization begins. The gating criterion is set to $\tau = 0.75$, and we use the same fixed threshold across all main experiments rather than tuning it separately for each optimization problem or LLM backbone. Vanilla BO and CAKE are initialized with random samples, while the remaining methods use the same LLM-suggested starting points. All results are averaged over five random seeds, with mean and standard deviation reported. For all LLM-assisted methods, we use **Intern S1 241B** (Bai et al., 2025), a large language model pretrained on scientific knowledge, which makes it well aligned with the prediction tasks considered in this work.

**Tasks.** We evaluate all methods on six formula optimization tasks spanning chemistry, materials science, energy systems, and nutrition. The dataset dimensions are indicated in parentheses:

**COF (14D)** (Gantzler et al., 2023): Maximize xenon/krypton adsorption selectivity by optimizing the pore structure and atomic composition of covalent organic frameworks.

*Table 1.* Ablation results on different gating criteria $\tau$ on COF (14D) and Fullerene (3D). We report the final objective, number of iterations to reach 90% of the best value, and the ratio of LLM- to real-fidelity queries (mean $\pm$ std over 5 runs).

| $\tau$ | COF | | | FULLERENE | | |
|---|---|---|---|---|---|---|
| | FINAL OBJ | ITERS TO 90% | L/R | FINAL OBJ | ITERS TO 90% | L/R |
| 0.60 | 10.778$\pm$0.276 | 24.60$\pm$2.51 | 1.52$\pm$0.29 | 0.9490$\pm$0.0019 | 22.40$\pm$6.35 | 1.54$\pm$0.23 |
| 0.65 | 10.934$\pm$0.165 | 21.60$\pm$5.81 | 1.91$\pm$0.94 | 0.9495$\pm$0.0015 | 22.20$\pm$2.86 | 1.85$\pm$0.52 |
| 0.70 | 11.070$\pm$0.224 | 15.83$\pm$7.78 | 2.00$\pm$0.34 | 0.9511$\pm$0.0014 | 17.00$\pm$9.17 | 2.00$\pm$0.73 |
| 0.75 | **11.228$\pm$0.162** | 14.17$\pm$6.62 | 2.68$\pm$1.13 | **0.9512$\pm$0.0012** | 14.60$\pm$5.68 | 3.87$\pm$1.62 |
| 0.80 | 11.134$\pm$0.156 | 14.80$\pm$5.26 | 3.44$\pm$1.22 | 0.9506$\pm$0.0020 | **14.50$\pm$8.58** | 5.69$\pm$2.44 |
| 0.85 | 11.171$\pm$0.121 | **12.60$\pm$7.77** | **5.26$\pm$2.11** | 0.9499$\pm$0.0010 | 15.00$\pm$2.35 | **14.60$\pm$3.70** |

**Fullerene (3D)** (Walker et al., 2017): Maximize the yield of C60 adducts in a cascaded reaction network by tuning time, temperature, and reagent concentration.

**PCE10 (4D)** (Langner et al., 2020): Minimize photodegradation in quaternary organic photovoltaic blends by adjusting the weight fractions of four polymer components.

**P3HT (5D)** (Bash et al., 2021): Maximize electrical conductivity in drop-cast P3HT/carbon nanotube composites by optimizing five continuous formulation variables.

**Flow Battery (3D)** (Zhou et al., 2024): Optimize a composite performance metric of a redox flow battery electrolyte by tuning the concentrations of ions.

**Sandwich (20D)** (Shams-White et al., 2023): Maximize a dietary health score over twenty ingredient quantities.

## 6.2. Main Results

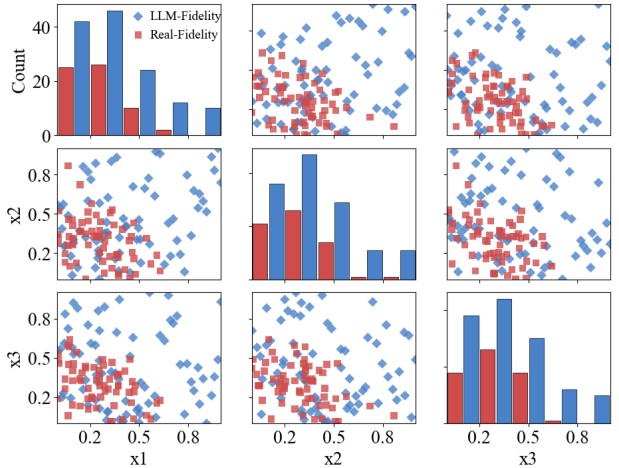

*Figure 4.* Distribution of LLM- and real-fidelity samples on the COF task. The first three normalized input dimensions are shown.

LABO consistently outperforms all baselines across the six optimization tasks (Figure 2). On relatively high-dimensional problems (COF and Sandwich), it achieves better final objective values within a limited evaluation bud-

get. While LLAMBO and BOPRO benefit from LLM suggested initialization with strong early progress, they tend to get stuck in local optima. In contrast, LABO leverages globally informed LLM-fidelity guidance to escape suboptimal regions and continue improving. On low-dimensional tasks like PCE10 and Flow Battery, where vanilla BO converges quickly, LABO still delivers better overall performance with more stable convergence. For more complex and noisy objectives such as Fullerene and P3HT, LABO again achieves the best performance. While CAKE demonstrates consistent exploratory behavior on these tasks, its performance tends to fluctuate, possibly due to instability introduced by adaptive kernel updates during optimization. In contrast, LABO exhibits significantly lower variance across random seeds, reflecting more consistent search behavior enabled by LLM-fidelity predictions grounded in prior knowledge and historical data.

Further comparison of sampling trajectories (Figure 3A) reveals that vanilla BO scatters its evaluations throughout the search process without a clear convergence pattern, whereas LABO rapidly focuses its samples on high-potential regions. This advantage stems from LABO's unique sampling mechanism. Taking the COF task as an example (Figure 4), LLM-fidelity queries broadly cover the input space, providing inexpensive, coarse global guidance, while real-fidelity evaluations are concentrated in a few sub-regions of high uncertainty, precisely as enforced by the gating criterion: expensive real-fidelity evaluations are triggered only when LLM predictions are deemed unreliable. As predicted by our theoretical analysis, this strategy drastically reduces the effective exploration region $\mathcal{X}_R^*$, leading to significantly improved sample efficiency.

## 6.3. Ablation Study

**LLM initialization.** To isolate the effect of LLM-suggested real-fidelity initialization, we initialize vanilla BO with the same set of candidates used to warm-start LABO. The LLM-fidelity exploratory phase remains unchanged. As shown in Figure 3B, LABO still significantly outperforms vanilla BO under identical initialization. This indicates that its

performance gain is not solely attributable to the initial real-fidelity candidates proposed by the LLM.

**Random-fidelity.** To assess the value of the LLM-fidelity signals, we replace them with uniformly sampled random values within the same output range. This leads to slower convergence and worse final performance (Figure 3B). This result confirms that LABO's advantage primarily arises from the informative predictions provided by the LLM, grounded in prior knowledge and in-context reasoning.

**Different LLMs.** We replace Intern S1 with a set of LLMs of varying sizes and capabilities, including Intern-S1-mini (7B) (Bai et al., 2025), Qwen3-235B-Instruct, Qwen3-235B-Thinking (Yang et al., 2025a), and Deepseek V3.1 (685B) (Liu et al., 2024a). The results (Figure 3C) show that the final optimization performance generally improves as model size increases. Qwen3-Thinking outperforms its Instruct counterpart in both convergence and final objective value, suggesting that models with enhanced reasoning capabilities can provide more reliable LLM-fidelity predictions. While stronger LLMs can further boost optimization performance, the overall differences across these models remain relatively modest. LABO maintains robust and effective performance across different types and capacities of LLMs.

**Effect of gating criterion.** We focus on varying the gating criterion $\tau$ from 0.60 to 0.85 because this range captures values where LABO strikes a balance between relying on LLM-fidelity predictions and real-fidelity evaluations. At lower values, LABO relies too heavily on real-fidelity evaluations, limiting the exploration benefits of LLM predictions. At higher values, LABO places too much trust in LLM outputs, which can mislead the search when the LLM predictions are inaccurate. As shown in Table 1, $\tau = 0.75$ yields the best performance on both tasks. As $\tau$ increases, LABO performs more LLM evaluations and develops a broader understanding of the search space, which can accelerate early-stage convergence. We also observe that the ratio of LLM- to real-fidelity queries is notably higher on the simpler Fullerene task, indicating greater reliance on LLM predictions. In contrast, high-dimensional tasks like COF require more real-fidelity queries to compensate for LLM uncertainty. These results suggest that LABO adjusts its fidelity allocation according to task complexity.

## 7. Conclusion

In conclusion, LABO effectively integrates LLMs into BO by exploiting their low evaluation cost for global guidance while reserving expensive real experiments for high-uncertainty regions. Theoretical analysis and extensive experiments demonstrate that LABO achieves superior sample efficiency and consistently outperforms existing approaches under practical experimental budgets.

## Acknowledgements

This work was supported by the New Generation Artificial Intelligence National Science and Technology Major Project (No. 2025ZD0121802). It was also supported by the New Generation Artificial Intelligence National Science and Technology Major Project (No. 2025ZD0122901) and the National Natural Science Foundation of China (No. 62572313).

## Impact Statement

This paper presents work whose goal is to advance the field of Machine Learning. There are many potential societal consequences of our work, none which we feel must be specifically highlighted here.

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

# Appendix

## A. Theorem

**Assumption A.1** (KOH-type joint GP). Let $\mathcal{X} \subset \mathbb{R}^d$ be compact. We assume that the LLM- and real-fidelity functions

$$f_L : \mathcal{X} \to \mathbb{R}, \qquad f_R : \mathcal{X} \to \mathbb{R}$$

form a zero-mean joint GP and a count $\rho$

$$f_L \sim \mathcal{GP}(0, k_L), \quad \delta \sim \mathcal{GP}(0, k_\delta), \quad f_R = \rho f_L + \delta$$

with co-kriging covariance structure: for any $x, x' \in \mathcal{X}$,

$$\text{Cov}(f_L(x), f_L(x')) = k_L(x, x'), \tag{2}$$
$$\text{Cov}(f_R(x), f_L(x')) = \rho\, k_L(x, x'), \tag{3}$$
$$\text{Cov}(f_R(x), f_R(x')) = \rho^2 k_L(x, x') + k_\delta(x, x'), \tag{4}$$

where $k_L$ and $k_\delta$ are positive definite kernels, and $\rho \in \mathbb{R}$ is a scalar scaling parameter.

**Lemma A.2.** *(Chowdhury & Gopalan, 2017, Theorem 2) Let $\mathcal{X}$ be compact and $k : \mathcal{X} \times \mathcal{X} \to \mathbb{R}$ a continuous kernel scaled so that $k(x, x) \leq 1$. Assume the (possibly mean-centered) target function $f$ satisfies $f \in \mathcal{H}_k$ with $\|f\|_{\mathcal{H}_k} \leq B$, and observations follow*

$$y_t = f(x_t) + \varepsilon_t, \qquad t = 1, 2, \ldots,$$

*where $\{\varepsilon_t\}$ are independent $R$-sub-Gaussian noise variables. Let $K_{t-1} = [k(x_i, x_j)]_{i,j=1}^{t-1}$, $k_{t-1}(x) = [k(x_1, x), \ldots, k(x_{t-1}, x)]^\top$, and define the GP posterior (with noise variance $\sigma^2$ and prior mean $0$) by*

$$\mu_{t-1}(x) = k_{t-1}(x)^\top (K_{t-1} + \sigma^2 I)^{-1} y_{1:t-1},$$

$$s_{t-1}^2(x) = k(x, x) - k_{t-1}(x)^\top (K_{t-1} + \sigma^2 I)^{-1} k_{t-1}(x),$$

*where $y_{1:t-1} = [y_1, \ldots, y_{t-1}]^\top$. Let the (maximal) information gain be*

$$\gamma_t := \max_{A \subset \mathcal{X},\, |A|=t} \tfrac{1}{2} \log \det(I + \sigma^{-2} K_A).$$

*Then for any $\delta \in (0, 1)$, with probability at least $1 - \delta$, simultaneously for all $t \geq 1$ and all $x \in \mathcal{X}$,*

$$\left| f(x) - \mu_{t-1}(x) \right| \leq \left( B + R\sqrt{2(\gamma_{t-1} + 1 + \ln \tfrac{1}{\delta})} \right) s_{t-1}(x).$$

*Equivalently, defining*

$$\beta_t(B) := B^2 + 2R^2(\gamma_{t-1} + 1 + \ln \tfrac{1}{\delta}),$$

*we have*

$$\left| f(x) - \mu_{t-1}(x) \right| \leq \sqrt{\beta_t(B)}\, s_{t-1}(x) \tag{5}$$

*uniformly over all $t, x$ with probability at least $1 - \delta$.*

**Lemma A.3.** *(Srinivas et al., 2009, Lemma 5.3 & Lemma 5.4) Under the same setting and notation as in Lemma A.2 (with $k(x, x) \leq 1$), for any chosen sequence $\{x_t\}_{t=1}^T$ the following hold:*

1. *(Information gain decomposition)*

$$I(y_{1:T}; f) = \tfrac{1}{2} \sum_{t=1}^T \log\left(1 + \sigma^{-2} s_{t-1}^2(x_t)\right).$$

2. *(Capped variance sum)*

$$\sum_{t=1}^T \min\left\{ 1,\, \sigma^{-2} s_{t-1}^2(x_t) \right\} \leq 2\gamma_T.$$

3. *(Width-sum bound) Consequently,*

$$\sum_{t=1}^{T} s_{t-1}(x_t) \leq \sqrt{2T\,\gamma_T}\,.\tag{6}$$

**Lemma A.4** (Equivalence to the KOH residual model). *Under Assumption A.1, Then:*

1. *$\delta$ is a GP with mean zero and covariance $k_\delta$:*
$$\delta \sim \mathcal{GP}(0, k_\delta).$$

2. *$\delta$ and $f_L$ are independent; i.e., for all $x, x' \in \mathcal{X}$,*
$$\mathrm{Cov}(\delta(x), f_L(x')) = 0.$$

3. *For all $x \in \mathcal{X}$, the variance decomposition holds:*
$$\mathrm{Var}(f_R(x)) = \rho^2 \mathrm{Var}(f_L(x)) + \mathrm{Var}(\delta(x)).$$

*Conversely, suppose $f_L \sim \mathcal{GP}(0, k_L)$ and $\delta \sim \mathcal{GP}(0, k_\delta)$ are independent GPs and define*

$$f_R(x) = \rho f_L(x) + \delta(x).$$

*Then $(f_L, f_R)$ is a joint GP satisfying the covariance structure in Assumption A.1.*

*Proof.* **(Forward direction).** Assume Assumption A.1 holds. Since $(f_L, f_R)$ is a joint GP, any linear transformation of $(f_L, f_R)$ is Gaussian. Hence the process

$$\delta(x) = f_R(x) - \rho f_L(x)$$

is Gaussian with mean zero.

We compute its covariance. For any $x, x' \in \mathcal{X}$,

$$\begin{aligned}
\mathrm{Cov}(\delta(x), \delta(x')) &= \mathrm{Cov}(f_R(x) - \rho f_L(x),\ f_R(x') - \rho f_L(x')) \\
&= \mathrm{Cov}(f_R(x), f_R(x')) - \rho\,\mathrm{Cov}(f_R(x), f_L(x')) \\
&\quad - \rho\,\mathrm{Cov}(f_L(x), f_R(x')) + \rho^2 \mathrm{Cov}(f_L(x), f_L(x')).
\end{aligned}$$

Substituting (2)–(4), we obtain

$$\mathrm{Cov}(\delta(x), \delta(x')) = k_\delta(x, x').$$

Therefore $\delta \sim \mathcal{GP}(0, k_\delta)$.

Next, for any $x, x'$,

$$\begin{aligned}
\mathrm{Cov}(\delta(x), f_L(x')) &= \mathrm{Cov}(f_R(x) - \rho f_L(x),\ f_L(x')) \\
&= \rho k_L(x, x') - \rho k_L(x, x') \\
&= 0.
\end{aligned}$$

Since the joint distribution is Gaussian, zero covariance implies independence. Thus $\delta \perp f_L$.

Finally, using $f_R = \rho f_L + \delta$, we have

$$\mathrm{Var}(f_R(x)) = \rho^2 \mathrm{Var}(f_L(x)) + \mathrm{Var}(\delta(x)) + 2\rho\,\mathrm{Cov}(f_L(x), \delta(x)).$$

The last term is zero, proving the variance decomposition.

**(Reverse direction).** Suppose instead that $f_L \sim \mathcal{GP}(0, k_L)$ and $\delta \sim \mathcal{GP}(0, k_\delta)$ are independent, and define $f_R = \rho f_L + \delta$. Then for any $x, x'$,

$$\mathrm{Cov}(f_R(x), f_L(x')) = \rho\, k_L(x, x'),$$

and

$$\mathrm{Cov}(f_R(x), f_R(x')) = \rho^2 k_L(x, x') + k_\delta(x, x').$$

Thus $(f_L, f_R)$ satisfies the covariance structure (2)–(4) in Assumption A.1. □

**Lemma A.5** (KOH posterior UCB confidence bound). *Suppose*

$$f_L \sim \mathcal{GP}(0, k_L), \qquad \delta \sim \mathcal{GP}(0, k_\delta),$$

*and $f_L \perp\!\!\!\perp \delta$. Let the real-fidelity function follow the Kennedy–O'Hagan autoregressive model*

$$f_R(x) = \rho\, f_L(x) + \delta(x),$$

*where $\rho$ is a fixed constant (within each iteration).*

*The real-fidelity observation model is*

$$y_t^H = f_R(x_t) + \varepsilon_t, \qquad t = 1, 2, \dots,$$

*where $\varepsilon_t$ are independent, zero-mean, R-sub-Gaussian noises:*

$$\mathbb{E}\big[e^{\lambda \varepsilon_t}\big] \le e^{\frac{\lambda^2 R^2}{2}}, \qquad \forall \lambda \in \mathbb{R}.$$

*Define the induced real-fidelity kernel*

$$k_R(x, x') := \rho^2 k_L(x, x') + k_\delta(x, x').$$

*Let $(\mu_{t-1}^H, \sigma_{t-1}^H)$ denote the GP posterior mean and standard deviation under prior kernel $k_R$ and observations $\mathcal{D}_{t-1} = \{(x_s, y_s^H)\}_{s=1}^{t-1}$.*

*Let $\gamma_t$ be the maximum information gain of kernel $k_R$ on $\mathcal{X}$:*

$$\gamma_t := \max_{A \subset \mathcal{X},\, |A|=t} I\big(f_R(A); y_A\big).$$

*Then there exists a universal constant $C > 0$ such that, for*

$$\beta_t = C\Big( \log(1/\delta) + \gamma_{t-1} \Big),$$

*the following high-probability event holds:*

$$\mathcal{E} := \Big\{ \big|f_R(x) - \mu_{t-1}^H(x)\big| \le \sqrt{\beta_t}\, \sigma_{t-1}^H(x), \quad \forall x \in \mathcal{X},\ \forall t \le T \Big\},$$

*and*

$$\mathbb{P}(\mathcal{E}) \ge 1 - \delta.$$

*Proof.* Since $f_L$ and $\delta$ are independent GPs, for any finite set $(x_1, \dots, x_n)$ we have

$$\mathbf{f}_L := (f_L(x_1), \dots, f_L(x_n))^\top \sim \mathcal{N}(0, K_L),$$

$$\boldsymbol{\delta} := (\delta(x_1), \dots, \delta(x_n))^\top \sim \mathcal{N}(0, K_\delta),$$

with $(\mathbf{f}_L, \boldsymbol{\delta})$ jointly Gaussian and independent. Thus

$$\mathbf{f}_R := (f_R(x_1), \dots, f_R(x_n))^\top = \rho \mathbf{f}_L + \boldsymbol{\delta} \sim \mathcal{N}\big(0,\ \rho^2 K_L + K_\delta\big).$$

Hence

$$\mathrm{Cov}(f_R(x_i), f_R(x_j)) = \rho^2 k_L(x_i, x_j) + k_\delta(x_i, x_j) = k_R(x_i, x_j),$$

so $f_R \sim \mathcal{GP}(0, k_R)$.

Given the observation model $y_t^H = f_R(x_t) + \varepsilon_t$ with sub-Gaussian noise, we are exactly in the setting of the GP-UCB theorem. By lemma A.2 choosing

$$\beta_t = C\big(\log(1/\delta) + \gamma_{t-1}\big)$$

ensures that the event

$$|f_R(x) - \mu_{t-1}^H(x)| \le \sqrt{\beta_t}\, \sigma_{t-1}^H(x), \qquad \forall x \in \mathcal{X},\ \forall t \le T,$$

holds simultaneously with probability at least $1 - \delta$. This coincides with the definition of $\mathcal{E}$.

$\square$

**Definition A.6** (Dynamic and limiting real-fidelity regions). At round $t$, let the KOH posterior variances be

$$\text{Var}_t[f_L(x)], \qquad \text{Var}_t[\delta(x)], \qquad \text{Var}_t[f_R(x)] := \rho^2 \text{Var}_t[f_L(x)] + \text{Var}_t[\delta(x)].$$

Define the *posterior mismatch ratio* at round $t$ as

$$p_\Delta^{(t)}(x) := \frac{\text{Var}_t[\delta(x)]}{\text{Var}_t[f_R(x)]} = \frac{\text{Var}_t[\delta(x)]}{\rho^2 \text{Var}_t[f_L(x)] + \text{Var}_t[\delta(x)]}.$$

Given a threshold $\alpha \in (0,1)$ and a cost ratio $c_L/c_R \in (0,1)$, the *dynamic real-fidelity region* at round $t$ is

$$\mathcal{X}_R^{(t)} := \left\{ x \in \mathcal{X} : p_\Delta^{(t)}(x) > \alpha \text{ or } p_\Delta^{(t)}(x) > 1 - \frac{c_L}{c_R} \right\}.$$

The *limiting real-fidelity region* is defined as the $\liminf$ of the dynamic regions:

$$\mathcal{X}_R^\star := \liminf_{t \to \infty} \mathcal{X}_R^{(t)} = \bigcup_{s=1}^\infty \bigcap_{t \geq s} \mathcal{X}_R^{(t)}.$$

We also define the limiting LLM-fidelity region

$$\mathcal{X}_L^\star := \mathcal{X} \setminus \mathcal{X}_R^\star.$$

**Lemma A.7** (Stabilization of the gate decisions). *Assume the setting of Lemma A.5 and Definition A.6, and assume the design space $\mathcal{X}$ is a finite set.[1]*

*For each $x \in \mathcal{X}$ and round $t \geq 0$, let $\text{Var}_t[f_L(x)]$, $\text{Var}_t[\delta(x)]$ and $\text{Var}_t[f_R(x)]$ denote the KOH posterior variances after $t$ rounds of (all-fidelity) observations, and define $p_\Delta^{(t)}(x)$, $\mathcal{X}_R^{(t)}$, $\mathcal{X}_R^\star$ and $\mathcal{X}_L^\star$ as in Definition A.6.*

*Assume that, for every $x \in \mathcal{X}$, the sequence $p_\Delta^{(t)}(x)$ converges to a limit*

$$p_\Delta^{(\infty)}(x) := \lim_{t \to \infty} p_\Delta^{(t)}(x),$$

*and that this limit is non-degenerate with respect to the gate thresholds:*

$$p_\Delta^{(\infty)}(x) \neq \alpha, \qquad p_\Delta^{(\infty)}(x) \neq 1 - \frac{c_L}{c_R}, \qquad \forall x \in \mathcal{X}.$$

*Then there exists a (possibly random but finite) time $T_0 < \infty$ such that, for all $t \geq T_0$ and all $x \in \mathcal{X}$,*

$$x \in \mathcal{X}_R^\star \iff x \in \mathcal{X}_R^{(t)}.$$

*Equivalently, for each $x \in \mathcal{X}$ there exists a finite time $T_x$ such that, for all $t \geq T_x$, the gate decision at $x$ does not change and agrees with its membership in the limiting region $\mathcal{X}_R^\star$.*

*Proof.* Fix any $x \in \mathcal{X}$. Since each GP posterior variance is obtained by conditioning on an increasing set of observations, the sequences $\text{Var}_t[f_L(x)]$ and $\text{Var}_t[\delta(x)]$ are non-increasing in $t$. They are also bounded below by 0, hence both sequences converge:

$$\text{Var}_t[f_L(x)] \to v_L(x) \geq 0, \qquad \text{Var}_t[\delta(x)] \to v_\delta(x) \geq 0.$$

Consequently,

$$\text{Var}_t[f_R(x)] = \rho^2 \text{Var}_t[f_L(x)] + \text{Var}_t[\delta(x)] \to v_R(x) := \rho^2 v_L(x) + v_\delta(x).$$

Since the map

$$(a,b) \mapsto \frac{b}{\rho^2 a + b}$$

---

[1] For continuous $\mathcal{X}$, one may apply a suitable $\varepsilon$-net discretization as in standard GP-UCB analyses; we present the finite case for clarity.

is continuous on $\{(a,b) \in [0,\infty)^2 : \rho^2 a + b > 0\}$, we obtain

$$p_\Delta^{(t)}(x) = \frac{\mathrm{Var}_t[\delta(x)]}{\mathrm{Var}_t[f_R(x)]} \longrightarrow p_\Delta^{(\infty)}(x) := \frac{v_\delta(x)}{\rho^2 v_L(x) + v_\delta(x)}.$$

Let the two thresholds be

$$\tau_1 := \alpha, \qquad \tau_2 := 1 - \frac{c_L}{c_R}.$$

By assumption, $p_\Delta^{(\infty)}(x) \neq \tau_1$ and $p_\Delta^{(\infty)}(x) \neq \tau_2$. Define the margin

$$\varepsilon_x := \tfrac{1}{2} \min\{|p_\Delta^{(\infty)}(x) - \tau_1|, |p_\Delta^{(\infty)}(x) - \tau_2|\} > 0.$$

Since $p_\Delta^{(t)}(x) \to p_\Delta^{(\infty)}(x)$, there exists $T_x < \infty$ such that

$$|p_\Delta^{(t)}(x) - p_\Delta^{(\infty)}(x)| \leq \varepsilon_x, \qquad \forall t \geq T_x.$$

If $x \in \mathcal{X}_R^\star$, then by definition

$$p_\Delta^{(\infty)}(x) > \tau_1 \quad \text{or} \quad p_\Delta^{(\infty)}(x) > \tau_2.$$

Without loss of generality, assume $p_\Delta^{(\infty)}(x) > \tau_1$. Then $p_\Delta^{(\infty)}(x) - \tau_1 \geq 2\varepsilon_x$, and hence

$$p_\Delta^{(t)}(x) \geq p_\Delta^{(\infty)}(x) - \varepsilon_x > \tau_1 \qquad (t \geq T_x),$$

so $x \in \mathcal{X}_R^{(t)}$ for all $t \geq T_x$.

If $x \in \mathcal{X}_L^\star$, then

$$p_\Delta^{(\infty)}(x) \leq \tau_1, \qquad p_\Delta^{(\infty)}(x) \leq \tau_2,$$

and at least one inequality is strict. Thus

$$\max\{p_\Delta^{(\infty)}(x) - \tau_1, \, p_\Delta^{(\infty)}(x) - \tau_2\} \leq -2\varepsilon_x,$$

so

$$p_\Delta^{(t)}(x) \leq p_\Delta^{(\infty)}(x) + \varepsilon_x < \max\{\tau_1, \tau_2\} \qquad (t \geq T_x),$$

and in particular

$$p_\Delta^{(t)}(x) \leq \tau_1, \qquad p_\Delta^{(t)}(x) \leq \tau_2.$$

Hence $x \notin \mathcal{X}_R^{(t)}$ for all $t \geq T_x$.

Thus for each $x \in \mathcal{X}$, its gate decision stabilizes at time $T_x$.

Since $\mathcal{X}$ is finite, define

$$T_0 := \max_{x \in \mathcal{X}} T_x < \infty.$$

Then for all $t \geq T_0$ and all $x \in \mathcal{X}$,

$$x \in \mathcal{X}_R^\star \quad \Longleftrightarrow \quad x \in \mathcal{X}_R^{(t)}.$$

Thus the dynamic real-fidelity region becomes constant for all $t \geq T_0$ and coincides with the limiting region $\mathcal{X}_R^\star$. The lemma is proved. $\qquad\square$

We define the instantaneous regret at round $t$ as

$$r_t := f - f_R(x_t), \quad f := \max_{x \in \mathcal{X}} f_R(x),$$

and decompose the index set $\{1, \dots, T\}$ according to the fidelity and the limiting regions:

$$\mathcal{T}_R^\star := \{t \leq T : m_t = H, \, x_t \in \mathcal{X}_R^\star\},$$

$$\mathcal{T}_R^{\mathrm{bad}} := \{t \le T : m_t = H,\ x_t \in \mathcal{X}_L^\star\}, \qquad \mathcal{T}_L := \{t \le T : m_t = L\}.$$

We also denote $T_R^\star := |\mathcal{T}_R^\star|$ and $T_L := |\mathcal{T}_L|$. Let $\Psi_T(\mathcal{A})$ denote the maximum mutual information of the real-fidelity GP with kernel $k_R$ over $T$ points chosen in $\mathcal{A} \subseteq \mathcal{X}$. We work on the high-probability event $\mathcal{E}$ of Lemma A.5 throughout the analysis.

**Lemma A.8** (Regret for real-fidelity pulls in the limiting HF region). *Work on the high-probability event $\mathcal{E}$ of Lemma A.5. Let $\mathcal{X}_R^\star$ be the limiting real-fidelity region from Definition A.6 and $\mathcal{T}_R^\star$ the set of rounds in which both $m_t = H$ and $x_t \in \mathcal{X}_R^\star$. Then there exists a constant $C > 0$ such that*

$$\sum_{t \in \mathcal{T}_R^\star} r_t \ \le\ C\sqrt{T_R^\star \beta_T \Psi_T(\mathcal{X}_R^\star)}.$$

*Proof.* On the event $\mathcal{E}$ of Lemma A.5, for any round $t$ and any $x \in \mathcal{X}$ we have the GP-UCB confidence bound

$$|f_R(x) - \mu_{t-1}^H(x)| \le \sqrt{\beta_t}\,\sigma_{t-1}^H(x).$$

Let $f = \max_{x \in \mathcal{X}} f_R(x)$ and $\mu_{t-1}^H$ be the posterior mean at round $t-1$. Then, as in the standard GP-UCB analysis,

$$\begin{aligned}
r_t &= f - f_R(x_t) \\
&\le \left(f - \mu_{t-1}^H(x_t)\right) + \left(\mu_{t-1}^H(x_t) - f_R(x_t)\right) \\
&\le \sqrt{\beta_t}\,\sigma_{t-1}^H(x_t) + \sqrt{\beta_t}\,\sigma_{t-1}^H(x_t) = 2\sqrt{\beta_t}\,\sigma_{t-1}^H(x_t),
\end{aligned}$$

for all $t \le T$.

Now restrict to $t \in \mathcal{T}_R^\star$. Since $\beta_t$ is non-decreasing in $t$, we have $\beta_t \le \beta_T$ for all $t \in \mathcal{T}_R^\star$, hence

$$\sum_{t \in \mathcal{T}_R^\star} r_t \le 2\sqrt{\beta_T} \sum_{t \in \mathcal{T}_R^\star} \sigma_{t-1}^H(x_t).$$

Next, we use the standard information-gain bound on the sum of posterior standard deviations. Let $A^\star := \{x_t : t \in \mathcal{T}_R^\star\} \subseteq \mathcal{X}_R^\star$ be the multiset of query points corresponding to real-fidelity pulls in the limiting region. Then, by Lemma A.3,

$$\sum_{t \in \mathcal{T}_R^\star} \sigma_{t-1}^H(x_t) \ \le\ C_1 \sqrt{T_R^\star \Psi_T(\mathcal{X}_R^\star)},$$

for some constant $C_1 > 0$, where $\Psi_T(\mathcal{X}_R^\star)$ is the maximum mutual information achievable by querying $T$ points in $\mathcal{X}_R^\star$ under the kernel $k_R$.

Combining the two inequalities yields

$$\sum_{t \in \mathcal{T}_R^\star} r_t \le 2\sqrt{\beta_T}\,C_1\sqrt{T_R^\star \Psi_T(\mathcal{X}_R^\star)} = C\sqrt{T_R^\star \beta_T \Psi_T(\mathcal{X}_R^\star)},$$

for an appropriate constant $C > 0$. This proves the lemma. $\qquad\square$

**Lemma A.9** (Regret outside $\mathcal{X}_R^\star$ and for LLM-fidelity rounds). *Work on the event $\mathcal{E}$ of Lemma A.5 and under the gate stabilization of Lemma A.7. Let*

$$\mathcal{T}_R^{\mathrm{bad}} := \{t \le T : m_t = H,\ x_t \in \mathcal{X}_L^\star\}, \qquad \mathcal{T}_L := \{t \le T : m_t = L\},$$

*and let $T_R^{\mathrm{bad}} := |\mathcal{T}_R^{\mathrm{bad}}|$ and $T_L := |\mathcal{T}_L|$.*

*Then, for any fixed $\alpha \in (0,1)$, there exist constants $C_2, C_3 > 0$ such that:*

1. *The number of real-fidelity pulls in the limiting LLM-fidelity region is sublinear:*

$$T_R^{\mathrm{bad}} \ \le\ C_2\,T^\alpha.$$

2. *The cumulative regret incurred by such real-fidelity pulls is bounded by*

$$\sum_{t \in \mathcal{T}_R^{\mathrm{bad}}} r_t \ \le\ C_3 \sqrt{T^\alpha \, \beta_T \, \Psi_T(\mathcal{X})}.$$

3. *The cumulative regret over all LLM-fidelity rounds satisfies*

$$\sum_{t \in \mathcal{T}_L} r_t \ \le\ C_3 \sqrt{T_L \, \beta_T \, \Psi_T(\mathcal{X})}.$$

*Proof.* **(i) Sublinear number of bad-region HF pulls.** By Lemma A.7, for every $x \in \mathcal{X}_L^\star$ there exists a finite time $T_x$ such that for all $t \ge T_x$, $x \notin \mathcal{X}_R^{(t)}$, i.e., the gate will always select low fidelity if the algorithm proposes to evaluate at $x$. Consequently, after time $T_x$ no further real-fidelity pulls can occur at $x$.

Moreover, on the event $\mathcal{E}$, the GP-UCB rule ensures that query points $x_t$ concentrate near the (unknown) maximizer, and in regions where the posterior variance is already small, the UCB value cannot remain large for many rounds. Combining these two facts yields a standard counting argument: for any fixed $\alpha \in (0, 1)$, there exists a constant $C_2 > 0$ such that

$$T_R^{\mathrm{bad}} = \left| \{ t \le T : m_t = H, \ x_t \in \mathcal{X}_L^\star \} \right| \ \le\ C_2 \, T^\alpha.$$

Intuitively, a real-fidelity pull in $\mathcal{X}_L^\star$ either occurs before the gate has stabilized at the corresponding point, or is triggered by an unusually large UCB value, both of which can only happen finitely or sublinearly often.

**(ii) Regret for bad-region HF pulls.** For $t \in \mathcal{T}_R^{\mathrm{bad}}$, we can reuse the UCB instantaneous regret bound on $\mathcal{E}$:

$$r_t \le 2\sqrt{\beta_t}\, \sigma_{t-1}^H(x_t) \le 2\sqrt{\beta_T}\, \sigma_{t-1}^H(x_t).$$

Summing over $t \in \mathcal{T}_R^{\mathrm{bad}}$,

$$\sum_{t \in \mathcal{T}_R^{\mathrm{bad}}} r_t \le 2\sqrt{\beta_T} \sum_{t \in \mathcal{T}_R^{\mathrm{bad}}} \sigma_{t-1}^H(x_t).$$

Let $A^{\mathrm{bad}} := \{ x_t : t \in \mathcal{T}_R^{\mathrm{bad}} \} \subseteq \mathcal{X}$ be the multiset of bad-region HF query points. Using the same information-gain argument as in Lemma A.8, but now over the full domain $\mathcal{X}$, we obtain

$$\sum_{t \in \mathcal{T}_R^{\mathrm{bad}}} \sigma_{t-1}^H(x_t) \ \le\ C_4 \sqrt{T_R^{\mathrm{bad}}\, \Psi_T(\mathcal{X})},$$

for some constant $C_4 > 0$. Together with $T_R^{\mathrm{bad}} \le C_2 T^\alpha$ this yields

$$\sum_{t \in \mathcal{T}_R^{\mathrm{bad}}} r_t \le 2\sqrt{\beta_T}\, C_4 \sqrt{T_R^{\mathrm{bad}}\, \Psi_T(\mathcal{X})} \le C_3 \sqrt{T^\alpha \, \beta_T \, \Psi_T(\mathcal{X})},$$

for an appropriate constant $C_3 > 0$.

**(iii) Regret for LLM-fidelity rounds.** For rounds $t \in \mathcal{T}_L$, the algorithm chooses the LLM-fidelity oracle. The instantaneous regret $r_t = f - f_R(x_t)$ is still defined with respect to the real-fidelity target $f_R$, but the query $(x_t, m_t = L)$ still contributes information to the KOH model: it reduces the posterior variance of $f_L$, and hence (through the relation $f_R = \rho f_L + \delta$) indirectly reduces $\mathrm{Var}_t[f_R]$ as well.

To bound the cumulative regret over $\mathcal{T}_L$, we again work on the event $\mathcal{E}$ and apply the UCB confidence interval at the proposed point $x_t$:

$$r_t = f - f_R(x_t) \le 2\sqrt{\beta_t}\, \sigma_{t-1}^H(x_t) \le 2\sqrt{\beta_T}\, \sigma_{t-1}^H(x_t),$$

for all $t \in \mathcal{T}_L$. Summing,

$$\sum_{t \in \mathcal{T}_L} r_t \le 2\sqrt{\beta_T} \sum_{t \in \mathcal{T}_L} \sigma_{t-1}^H(x_t).$$

Let $A_L := \{x_t : t \in \mathcal{T}_L\} \subseteq \mathcal{X}$ be the multiset of LLM-fidelity query points. The same information-gain argument applied to these $T_L$ rounds (again over the full domain $\mathcal{X}$) yields

$$\sum_{t \in \mathcal{T}_L} \sigma_{t-1}^H(x_t) \;\le\; C_4 \sqrt{T_L \, \Psi_T(\mathcal{X})}.$$

Therefore

$$\sum_{t \in \mathcal{T}_L} r_t \le 2\sqrt{\beta_T}\, C_4 \sqrt{T_L \, \Psi_T(\mathcal{X})} \le C_3 \sqrt{T_L \, \beta_T \, \Psi_T(\mathcal{X})},$$

for a (possibly larger) constant $C_3 > 0$. This completes the proof. $\qquad\square$

**Theorem A.10** (Regret of multi-fidelity BO with KOH model and gating). *Assume the Kennedy–O'Hagan model*

$$f_L \sim \mathcal{GP}(0, k_L), \qquad \delta \sim \mathcal{GP}(0, k_\delta), \qquad f_R(x) = \rho\, f_L(x) + \delta(x),$$

*with $f_L \perp\!\!\!\perp \delta$ and fixed $\rho$, and the real-fidelity observation model*

$$y_t^H = f_R(x_t) + \varepsilon_t,$$

*where $(\varepsilon_t)_t$ are independent, zero-mean, R-sub-Gaussian noises. Let $k_R(x, x') := \rho^2 k_L(x, x') + k_\delta(x, x')$ denote the induced real-fidelity kernel.*

*At each round $t$, let $(\mu_{t-1}^H, \sigma_{t-1}^H)$ be the GP posterior under $k_R$ conditioned on all past observations, and let the candidate point be chosen by the real-fidelity UCB rule*

$$x_t \in \arg\max_{x \in \mathcal{X}} \Big\{ \mu_{t-1}^H(x) + \sqrt{\beta_t}\, \sigma_{t-1}^H(x) \Big\},$$

*with*

$$\beta_t = C_0 \big( \log(1/\delta) + \gamma_{t-1} \big),$$

*for a suitable constant $C_0 > 0$, where $\gamma_t$ is the maximum information gain associated with $k_R$.*

*The fidelity $m_t \in \{L, H\}$ is then selected by the two-gate policy based on the mismatch ratio*

$$p_\Delta^{(t)}(x) = \frac{\mathrm{Var}_t[\delta(x)]}{\mathrm{Var}_t[f_R(x)]} = \frac{\mathrm{Var}_t[\delta(x)]}{\rho^2\, \mathrm{Var}_t[f_L(x)] + \mathrm{Var}_t[\delta(x)]},$$

*namely,*

$$m_t = H \quad \Longleftrightarrow \quad p_\Delta^{(t)}(x_t) > \alpha \ \text{or} \ p_\Delta^{(t)}(x_t) > 1 - \frac{c_L}{c_R},$$

*for fixed $\alpha \in (0, 1)$ and cost ratio $c_L/c_R \in (0, 1)$; otherwise $m_t = L$. Let the dynamic and limiting real-fidelity regions $\mathcal{X}_R^{(t)}$ and $\mathcal{X}_R^\star$ be defined as in Definition A.6, and assume the gate stabilization conditions of Lemma A.7 hold.*

*Define the instantaneous regret*

$$r_t := f - f_R(x_t), \quad f := \max_{x \in \mathcal{X}} f_R(x),$$

*and the cumulative regret $R_T := \sum_{t=1}^T r_t$. Decompose the index sets as*

$$\mathcal{T}_R^\star := \{t \le T : m_t = H, \ x_t \in \mathcal{X}_R^\star\},$$

$$\mathcal{T}_R^{\mathrm{bad}} := \{t \le T : m_t = H, \ x_t \in \mathcal{X}_L^\star\}, \qquad \mathcal{T}_L := \{t \le T : m_t = L\},$$

*and denote $T_R^\star := |\mathcal{T}_R^\star|$ and $T_L := |\mathcal{T}_L|$. Let $\Psi_T(\mathcal{A})$ be the maximum mutual information of the real-fidelity GP with kernel $k_R$ over $T$ points in $\mathcal{A} \subseteq \mathcal{X}$.*

*Then, for any $\delta \in (0, 1)$ and any fixed $\alpha \in (0, 1)$, there exists a constant $C > 0$ such that, with probability at least $1 - \delta$,*

$$R_T \;\le\; C_1 \sqrt{T_R^\star \, \beta_T \, \Psi_T(\mathcal{X}_R^\star)} \;+\; C_2 \sqrt{T^\alpha \, \beta_T \, \Psi_T(\mathcal{X})} \;+\; C_3 \sqrt{T_L \, \beta_T \, \Psi_T(\mathcal{X})}.$$

*Proof.* We work on the high-probability event $\mathcal{E}$ of Lemma A.5, on which the GP-UCB confidence bounds hold for all $x \in \mathcal{X}$ and all $t \leq T$.

By Definition A.6 and Lemma A.7, the dynamic real-fidelity regions $\mathcal{X}_R^{(t)}$ stabilize to the limiting region $\mathcal{X}_R^{\star}$ after a finite time. We decompose the index set $\{1, \dots, T\}$ into three disjoint subsets:

$$\mathcal{T}_R^{\star}, \quad \mathcal{T}_R^{\text{bad}}, \quad \mathcal{T}_L,$$

corresponding respectively to real-fidelity pulls in $\mathcal{X}_R^{\star}$, real-fidelity pulls in $\mathcal{X}_L^{\star}$, and LLM-fidelity rounds.

Lemma A.8 shows that, on $\mathcal{E}$,

$$\sum_{t \in \mathcal{T}_R^{\star}} r_t \ \leq \ C_1 \sqrt{T_R^{\star} \, \beta_T \, \Psi_T(\mathcal{X}_R^{\star})}.$$

Lemma A.9 establishes that, for any fixed $\alpha \in (0, 1)$,

$$T_R^{\text{bad}} \leq C_2 T^{\alpha}, \qquad \sum_{t \in \mathcal{T}_R^{\text{bad}}} r_t \ \leq \ C_3 \sqrt{T^{\alpha} \, \beta_T \, \Psi_T(\mathcal{X})},$$

and

$$\sum_{t \in \mathcal{T}_L} r_t \ \leq \ C_3 \sqrt{T_L \, \beta_T \, \Psi_T(\mathcal{X})}.$$

Summing these three contributions and absorbing constants yields the stated bound on $R_T$. Using $T_R^{\star} \leq T$ and $T_L \leq T$ and the standard growth rates of $\beta_T$ and $\Psi_T(\cdot)$ gives the simplified $\tilde{\mathcal{O}}(\cdot)$ form. $\qquad \square$

**Remark A.11** (Comparison with MF-GP-UCB). Classical MF-GP-UCB relies on the assumption that the LLM-fidelity function is uniformly informative through the $\zeta$-bounded difference condition. Such an assumption often requires a very large $\zeta$, making the bound extremely loose, and is thus frequently violated in practice. In modern settings where the LLM-fidelity source is a large language model: its predictions may exhibit large and highly nonstationary variance across the domain. In contrast, our gating criterion $p_\Delta^{(t)}(x)$ identifies regions where the KOH residual dominates and automatically isolates a stable real-fidelity region $\mathcal{X}_R^{\star}$. As a consequence, the algorithm does not rely on a global fidelity ordering and adapts seamlessly to high-variance, heterogeneous fidelity sources such as LLMs.

**Remark A.12** (Advantage over standard GP-UCB). Standard GP-UCB explores the entire domain under a single real-fidelity model and therefore suffers regret proportional to the information capacity $\Psi_T(\mathcal{X})$ of the full search space. In our method, the gating mechanism rapidly filters out regions where LLM-fidelity signals are uninformative and where real-fidelity evaluations offer no meaningful variance reduction. This yields a significantly reduced effective search region $\mathcal{X}_R^{\star}$, leading to regret controlled by $\Psi_T(\mathcal{X}_R^{\star}) \ll \Psi_T(\mathcal{X})$ and dramatically shrinking exploration in suboptimal or "bad" areas.

# B. Prompt

To ensure reproducibility, transparency, and adaptability, we design the prompt for LABO in a modular fashion, consisting of a **fixed system prompt** and a **dynamic user prompt**. The **system prompt** is invariant across tasks and encodes the core scientific principles and output rules that the LLM must follow, while the **user prompt** is flexible and task-specific, providing experiment-specific background and historical context. This separation allows LABO to (i) maintain consistent reasoning discipline, (ii) incorporate domain knowledge explicitly, and (iii) adapt flexibly to different experimental datasets.

### System Prompt

The **system prompt** is fixed and instructs the LLM to respond strictly in valid JSON format, without any additional commentary or markdown. The format of the response is rigidly defined to ensure consistency across tasks.

```
SYSTEM_PROMPT = """You are a scientific assistant supporting Bayesian Optimization studies
    .
- Output ONLY valid JSON. No markdown, no explanations, no comments, no thinking process.
- Respond immediately with the required JSON format.
- Always keep feature order as provided and clip predictions to their valid numeric bounds
    .
- Treat historical measurements as read-only context; never copy them back into the output
    ."""
```

**User Prompt**

The **user prompt** is dynamically structured and provides the LLM with the necessary background, parameters, goal, and historical data. The LLM is instructed to predict future data points based on the current context. The output format is strict, and predictions must be returned in the specified JSON schema.

```
Background:
We are conducting a Bayesian Optimization task to optimize a set of experimental
    conditions. The goal is to maximize a certain outcome, based on the observed
    relationship between input features and the target. The optimization is guided by a
    surrogate model that balances exploration and exploitation based on previous
    observations. The task involves selecting new points to evaluate based on the
    surrogate model's predictions.

Parameters:
- Experimental Conditions: [x1, x2, x3] (These represent the input features or parameters
    that influence the outcome of the experiments)
- Objective Function: The objective is to predict the outcome (y) based on the above
    conditions.

Goal:
Maximize the target outcome by optimizing the experimental conditions. The objective
    function is costly to evaluate, and the aim is to focus on areas with high potential,
    based on prior data and the surrogate model's predictions.

Historical Data (read-only context):
{
    "data_points": [
        {
            "features": [0.1, 0.2, 0.3],
            "target": 0.95
        },
        {
            "features": [0.4, 0.5, 0.6],
            "target": 0.87
        },
        ...
    ]
}

Predict These Points (return predictions in the same order):
{
    "data_points": [
        {
            "features": [x1, x2, x3]
        },
        ...
    ]
}

REQUIRED OUTPUT FORMAT (exact schema):
{
    "data_points": [
        {
            "features": [x1, x2, x3],
            "target": y
        },
        {
            "features": [x1, x2, x3],
            "target": y
        }
    ]
}

CRITICAL FORMATTING REQUIREMENTS:
```

```
1. Output ONLY the JSON object. No text before or after. No explanations. No markdown
   formatting.
2. Response must be valid JSON with double quotes, no comments, and no trailing commas.
3. The root object must contain exactly one key: "data_points" (an array).
4. The "data_points" array must contain exactly one entry for each input point, in the
   same order as provided.
5. Each entry must be an object with exactly two keys:
   - "features": an array of numbers in the same order as the provided input features
   - "target": a single number (the predicted f value, rounded to 3 decimals, clipped to [
     $y_min$,$y_max$])
6. Do not include explanations, markdown, historical records, or any other keys or text
   outside the JSON object.
7. The number of entries in "data_points" must exactly match the number of input points
   provided.
```

This format ensures a clear, structured approach to designing prompts for LABO, maintaining both consistency and flexibility across different experimental setups. The system prompt ensures consistency in output, while the user prompt adapts to the specifics of the optimization task, allowing LABO to efficiently guide BO using the LLM.

## C. Experiments

### C.1. Datasets

This section provides detailed descriptions of the six experiments used in this study. Each task focuses on the optimization of a particular system or material, with the goal of improving performance based on various parameters.

**Sandwich (Shams-White et al., 2023).** This task aims to optimize the ingredient quantities for a sandwich, where the goal is to maximize the **Total_Score**, which combines two components: **Total_HEI_Score** and **Calorie_Score**. The **Total_HEI_Score** evaluates nutritional quality based on the HEI-2020 system, and the **Calorie_Score** rewards the sandwich's energy content being near 1200 kcal. The optimization involves balancing the adequacy and moderation components, considering the trade-offs between ingredients that contribute positively or negatively to the HEI and calorie targets.

| Input/Output | Variable Name | Range | Description | Type |
|---|---|---|---|---|
| **Inputs** | multigrain_bread | [0.0, 140.0] | Amount of multigrain bread (g) | Continuous |
| | whole_wheat_bread | [0.0, 140.0] | Amount of whole wheat bread (g) | Continuous |
| | sourdough_bread | [0.0, 140.0] | Amount of sourdough bread (g) | Continuous |
| | chicken_protein | [0.0, 100.0] | Amount of chicken protein (g) | Continuous |
| | tuna_protein | [0.0, 100.0] | Amount of tuna protein (g) | Continuous |
| | tofu_protein | [0.0, 80.0] | Amount of tofu protein (g) | Continuous |
| | hummus_protein | [0.0, 70.0] | Amount of hummus protein (g) | Continuous |
| | egg_protein | [0.0, 80.0] | Amount of egg protein (g) | Continuous |
| | low_fat_cheese_dairy | [0.0, 20.0] | Amount of Low-fat cheese dairy (g) | Continuous |
| | cheddar_cheese | [0.0, 20.0] | Amount of cheddar cheese (g) | Continuous |
| | swiss_cheese_dairy | [0.0, 20.0] | Amount of Swiss cheese dairy (g) | Continuous |
| | collards | [0.0, 80.0] | Amount of collards (g) | Continuous |
| | cabbage | [0.0, 80.0] | Amount of cabbage (g) | Continuous |
| | onion_vegetables | [0.0, 80.0] | Amount of onion vegetables (g) | Continuous |
| | tomato_vegetables | [0.0, 80.0] | Amount of tomato vegetables (g) | Continuous |
| | mayo_sauce | [0.0, 15.0] | Amount of mayonnaise sauce (g) | Continuous |
| | olive_oil | [0.0, 20.0] | Amount of olive oil (g) | Continuous |
| | apples | [0.0, 100.0] | Amount of apples (g) | Continuous |
| | orange | [0.0, 100.0] | Amount of orange (g) | Continuous |
| | banana | [0.0, 100.0] | Amount of banana (g) | Continuous |
| **Output** | Total_Score | [0, 200] | Score combining HEI and Calorie scores | Continuous |

*Table 2.* Input and Output variables for Sandwich task.

**Fullerene (Walker et al., 2017).** This task optimizes a cascadic reaction forming o-xylenyl adducts of C60. The goal is to maximize the **mole_fraction** of useful adducts (X1 + X2), which directly impacts product selectivity and yield. The task

involves tuning three key reaction parameters: reaction time, sultine concentration, and temperature. Parameters are all real-valued in physical units (not normalized). The dataset used for optimization includes 246 sets of data, which provide a strong dataset to guide predictions, exhibiting nonlinear dependencies on reaction conditions.

| Input/Output | Variable Name | Range | Description | Type |
|---|---|---|---|---|
| **Inputs** | reaction_time | [3, 31] (min) | Duration of the reaction (minutes) | Continuous |
| | sultine_conc | [1.5, 6.0] | Concentration of sultine reactant | Continuous |
| | temperature | [100, 150] (°C) | Temperature | Continuous |
| **Output** | mole_fraction | [0.0, 1.0] | The mole fraction of useful adducts | Continuous |

*Table 3.* Input and Output variables for Fullerene task.

**PCE10 (Langner et al., 2020).** This task optimizes the composition of quaternary organic photovoltaic (OPV) active-layer blends to minimize photodegradation, a critical metric that measures photostability after light exposure. The goal is to minimize degradation, which directly influences device lifetime and performance retention. The task involves tuning four key compositional parameters: PCE10, P3HT, PCBM, and olDTBR. The dataset used for this task includes 1040 measurements and exhibits strong nonlinear dependencies between the parameters, with degradation being dominated by acceptor-acceptor interactions rather than simple linear effects.

| Input/Output | Variable Name | Range | Description | Type |
|---|---|---|---|---|
| **Inputs** | PCE10 | [0.00, 1.00] | Weight fraction of PCE10 polymer donor | Continuous |
| | P3HT | [0.00, 1.00] | Weight fraction of P3HT polymer donor | Continuous |
| | PCBM | [0.00, 1.00] | Weight fraction of PCBM fullerene acceptor | Continuous |
| | olDTBR | [0.00, 1.00] | Weight fraction of olDTBR non-fullerene acceptor | Continuous |
| **Output** | degradation | [0.0, 0.75] | The degradation metric (lower is better) | Continuous |

*Table 4.* Input and Output variables for PCE10 task.

**COF (Gantzler et al., 2023).** This task optimizes Covalent Organic Framework materials for efficient Xe/Kr gas separation by tuning 14 key structural and compositional parameters. The objective is to maximize gcmc_y, the Xe/Kr adsorption selectivity derived from Grand Canonical Monte Carlo (GCMC) real-fidelity simulations. A gcmc_y value greater than 1 indicates that the material is more selective for Xe over Kr, with higher values indicating superior separation performance. The dataset used for this task contains 608 entries, with data showing that smaller pores, lower void fraction, and higher crystalline density are associated with higher gcmc_y values.

| Input/Output | Variable Name | Range | Description | Type |
|---|---|---|---|---|
| **Inputs** | pore_diameter | [3.5094, 56.3986] (Å) | Equivalent pore diameter (Å) | Continuous |
| | void_fraction | [0.1636, 0.9278] | Porosity | Continuous |
| | surface_area | [1996.63, 6357.01] (m²/g) | Specific surface area (m²/g) | Continuous |
| | crystal_density | [102.7248, 1610.7018] (kg/m³) | Crystalline density (kg/m³) | Continuous |
| | B | [0.0000, 0.1818] (atomic fraction) | Boron atomic fraction | Continuous |
| | O | [0.0000, 0.2500] (atomic fraction) | Oxygen atomic fraction | Continuous |
| | C | [0.3250, 0.6667] (atomic fraction) | Carbon atomic fraction | Continuous |
| | H | [0.0000, 0.5000] (atomic fraction) | Hydrogen atomic fraction | Continuous |
| | Si | [0.0000, 0.0295] (atomic fraction) | Silicon atomic fraction | Continuous |
| | N | [0.0000, 0.3333] (atomic fraction) | Nitrogen atomic fraction | Continuous |
| | S | [0.0000, 0.1429] (atomic fraction) | Sulfur atomic fraction | Continuous |
| | P | [0.0000, 0.0667] (atomic fraction) | Phosphorus atomic fraction | Continuous |
| | halogens | [0.0000, 0.2857] (atomic fraction) | Combined halogens | Continuous |
| | metals | [0.0000, 0.0238] (atomic fraction) | Combined metal content | Continuous |
| **Output** | gcmc_y | [0.0, 5.0] | Xe/Kr adsorption selectivity | Continuous |

*Table 5.* Input and Output variables for COF task.

**Flow Battery (Zhou et al., 2024).** This task optimizes the composition of an iron-chromium liquid flow battery with 4000 water molecules as the base. The parameters Fe, Cr, and H represent the number of ions added to this aqueous base. The objective is to maximize the comprehensive performance indicator $f$, which balances conductivity, viscosity, and ionic

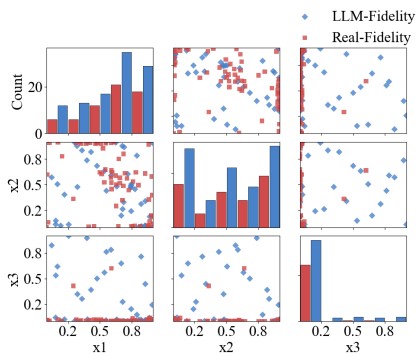 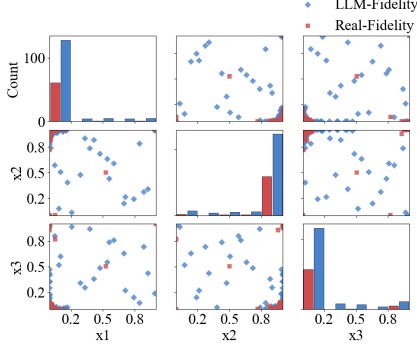

*Figure 5.* Distribution of LLM- and real-fidelity samples on the Fullerene and flowbattery task. The first three normalized input dimensions are shown.

concentration effects. The optimization must carefully manage trade-offs: higher H and lower Fe can boost conductivity, but may hurt the overall balance, while lower Cr can reduce viscosity but risks underperformance at very low Cr levels. The dataset contains 615 data points, which will be used to guide the optimization process.

| Input/Output | Variable Name | Range | Description | Type |
|---|---|---|---|---|
| **Inputs** | Fe_particle_number | [50, 145] (integers) | Number of Fe particles | Discrete (Integer) |
| | Cr_particle_number | [55, 145] (integers) | Number of Cr particles | Discrete (Integer) |
| | H_particle_number | [109, 289] (integers) | Number of H particles | Discrete (Integer) |
| **Output** | performance_indicator_f | [0, 1] | Performance indicator $f$ | Continuous |

*Table 6.* Input and Output variables for Flow Battery task.

**P3HT** (Bash et al., 2021). This task optimizes the composition of drop-cast P3HT/CNT composite thin films for electrical conductivity by tuning five key compositional parameters. The objective is to maximize **electrical conductivity**, a critical metric for electronic device performance that directly influences charge transport efficiency and device functionality. Data-driven guidance from this dataset suggests that the highest conductivity occurs with specific combinations of the polymer donor (P3HT) and carbon nanotubes (CNTs). The dataset used for this task contains 233 data points, which will guide the optimization process by exploring various compositions and their effects on conductivity.

| Input/Output | Variable Name | Range | Description | Type |
|---|---|---|---|---|
| **Inputs** | p3ht_content | [10, 100] (%) | rr-P3HT polymer content in composite film | Continuous |
| | d1_content | [0, 60] (%) | l-SWNT carbon nanotube content | Continuous |
| | d2_content | [0, 70] (%) | s-SWNT carbon nanotube content | Continuous |
| | d6_content | [0, 85] (%) | MWCNT carbon nanotube content | Continuous |
| | d8_content | [0, 75] (%) | DWCNT carbon nanotube content | Continuous |
| **Output** | electrical_conductivity | [0, 1] | Electrical conductivity | Continuous |

*Table 7.* Input and Output variables for P3HT task.

### C.2. More Results

In this section we provide more scatter diagrams in the following datasets. As shown in Figure 5–7, LLM-fidelity queries cover the input space broadly, while real-fidelity evaluations are concentrated in a few subregions. This allocation reflects LABO's design: a global LLM-fidelity model provides inexpensive, coarse guidance, while a discrepancy correction targets areas with high uncertainty. The gating criterion enforces this separation, focusing real-fidelity queries where LLM predictions lack reliability. As predicted by our theoretical analysis, this leads to a much smaller effective exploration region $\mathcal{X}_R^*$ and improved sample efficiency.

### C.3. Additional Experimental Results

In this section, we report additional experimental results that evaluate LABO from several complementary perspectives, including AutoML hyperparameter optimization, high-dimensional scientific optimization, comparison with LLM-based

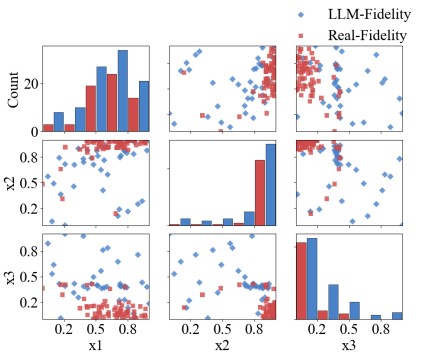 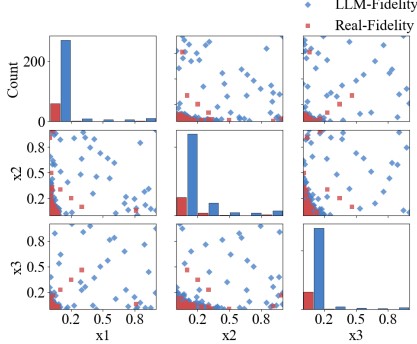

*Figure 6.* Distribution of LLM- and real-fidelity samples on the P3HT and PCE10 task. The first three normalized input dimensions are shown.

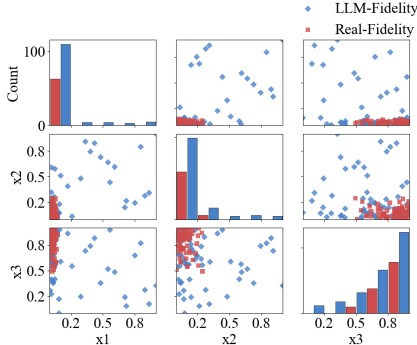 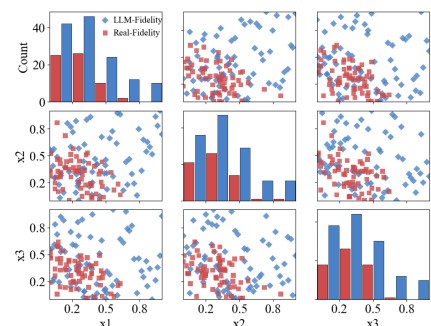

*Figure 7.* Distribution of LLM- and real-fidelity samples on the Sandwich and COF task. The first three normalized input dimensions are shown.

Thompson sampling, cost-benefit analysis, and empirical alignment between LLM-fidelity and real-fidelity observations.

### C.3.1. AUTOML HYPERPARAMETER OPTIMIZATION

To evaluate whether LABO is also effective beyond scientific formulation tasks, we add two AutoML hyperparameter optimization benchmarks from HPOBench (Eggensperger et al., 2021): SVM and MLP. The SVM task has a 2-dimensional search space and the MLP task has a 5-dimensional search space. Both tasks are evaluated for 80 optimization iterations. As shown in Table 8, LABO achieves the best final objective value and the fastest convergence on both tasks, demonstrating that the proposed LLM-fidelity guidance and gating mechanism are also effective for machine learning optimization problems.

*Table 8.* Additional AutoML results on HPOBench. We report the final objective value and the number of iterations required to reach 90% of the best value. Results are reported as mean $\pm$ standard deviation over five runs.

| Method | Final Obj. ↑ (SVM) | Iters to 90% ↓ (SVM) | Final Obj. ↑ (MLP) | Iters to 90% ↓ (MLP) |
|---|---|---|---|---|
| LABO | **0.909 ± 0.016** | **33.8 ± 18.8** | **0.941 ± 0.009** | **30.0 ± 10.2** |
| LLAMBO | 0.892 ± 0.023 | 42.0 ± 14.5 | 0.925 ± 0.006 | 50.2 ± 20.8 |
| BOPRO | 0.898 ± 0.019 | 60.2 ± 9.6 | 0.934 ± 0.013 | 47.2 ± 24.9 |
| CAKE | 0.904 ± 0.014 | 50.0 ± 24.5 | 0.939 ± 0.016 | 52.2 ± 15.0 |
| Vanilla BO | 0.880 ± 0.018 | 42.8 ± 20.1 | 0.905 ± 0.014 | 54.4 ± 16.8 |

### C.3.2. HIGH-DIMENSIONAL SCIENTIFIC OPTIMIZATION

We further evaluate LABO on an 86-dimensional superconductor optimization task (Hamidieh, 2018), where the goal is to maximize the superconducting critical temperature. This task provides a substantially higher-dimensional scientific optimization benchmark than those used in the main experiments. All methods are evaluated for 80 optimization iterations. As shown in Table 9, LABO substantially outperforms all baselines in both final objective value and convergence speed. This result indicates that LABO remains effective in high-dimensional scientific optimization settings.

*Table 9.* Additional results on the 86D superconductor task. We report the final objective value and the number of iterations required to reach 90% of the best value. Results are reported as mean $\pm$ standard deviation over five runs.

| Method | Final Obj. $\uparrow$ | Iters to 90% $\downarrow$ |
| --- | --- | --- |
| LABO | **91.06 $\pm$ 8.05** | **24.4 $\pm$ 23.3** |
| LLAMBO | 72.20 $\pm$ 19.78 | 51.2 $\pm$ 26.9 |
| BOPRO | 80.38 $\pm$ 8.37 | 34.6 $\pm$ 31.7 |
| CAKE | 76.57 $\pm$ 10.82 | 36.6 $\pm$ 26.6 |
| Vanilla BO | 57.96 $\pm$ 18.33 | 64.6 $\pm$ 14.3 |

### C.3.3. COMPARISON WITH DIRECT LLM-BASED THOMPSON SAMPLING

Recent LLM-based BO methods have explored using LLMs more directly in the sampling process. To further position LABO with respect to this line of work, we compare LABO with ToSFiT (Menet et al., 2025), which reformulates Thompson sampling as an online LLM fine-tuning process for discrete optimization. We conduct the comparison on a discrete CBD lipid nanoparticle formulation task (Xie et al., 2024). As shown in Table 10, LABO achieves both a better final objective value and faster convergence than ToSFiT and vanilla BO. This suggests that LABO's fidelity-aware integration of LLM predictions provides an effective alternative to directly using LLMs as the sampling mechanism.

*Table 10.* Comparison with ToSFiT on the CBD lipid nanoparticle formulation task. We report the final objective value and the number of iterations required to reach 90% of the best value. Results are reported as mean $\pm$ standard deviation over five runs.

| Method | Final Obj. $\uparrow$ | Iters to 90% $\downarrow$ |
| --- | --- | --- |
| LABO | **0.898 $\pm$ 0.017** | **4.4 $\pm$ 3.4** |
| ToSFiT | 0.883 $\pm$ 0.017 | 6.0 $\pm$ 5.5 |
| Vanilla BO | 0.711 $\pm$ 0.016 | 6.4 $\pm$ 2.8 |

### C.3.4. COST-BENEFIT ANALYSIS

Since LABO uses additional LLM queries to reduce the number of expensive real-fidelity evaluations, we further compare the total cost required by LABO and vanilla BO to reach the same performance level. On the COF task, we consider a conservative cost setting where each LLM query costs \$1 and each real-fidelity experiment costs \$100. We compare the total cost needed to reach the final performance achieved by vanilla BO. As shown in Table 11, vanilla BO requires 30 rounds and 60 real-fidelity evaluations, resulting in a total cost of \$6000. In contrast, LABO reaches the same performance level within 4 rounds, using 164 LLM queries and 8 real-fidelity evaluations, resulting in a total cost of \$964. This demonstrates that LABO can substantially reduce the overall optimization cost when real-fidelity evaluations are expensive.

*Table 11.* Cost-benefit comparison on the COF task. We compare the total cost required to reach the final performance level achieved by vanilla BO. The cost is computed under a conservative setting where each LLM query costs \$1 and each real-fidelity experiment costs \$100.

| Method | Rounds | LLM Queries | Total Cost (\$) |
| --- | --- | --- | --- |
| Vanilla BO | 30 | 0 | 6000 |
| LABO | 4 | 164 | 964 |

### C.3.5. ALIGNMENT BETWEEN LLM-FIDELITY AND REAL-FIDELITY OBSERVATIONS

LABO models the real-fidelity function using the Kennedy–O'Hagan decomposition

$$f_R(x) = \rho f_L(x) + \delta(x), \tag{7}$$

where $\rho$ captures global scale alignment and $\delta(x)$ models the remaining discrepancy between LLM-fidelity and real-fidelity observations. To empirically examine whether LLM-fidelity predictions provide useful signal, we measure the correlation between LLM-fidelity and real-fidelity observations on the Flow Battery task. The LLM-fidelity predictions achieve $R^2 = 0.649$ and Pearson correlation $r = 0.806$ with real-fidelity observations, indicating a strong positive association. This supports the use of LLM predictions as informative low-fidelity signals in LABO. When the LLM signal is less informative

or systematically misleading, the discrepancy term becomes dominant and the gating criterion causes LABO to rely more on real-fidelity evaluations, as also reflected by the random-fidelity ablation in Section C.

