# OpenReview forum: "LABO: LLM-Accelerated Bayesian Optimization through Broad Exploration and Selective Experimentation"
_ICML.cc/2026/Conference — ICML 2026 regular_

### Official Review · Reviewer_y8XY · 2026-02-22

**Soundness:** 3
**Presentation:** 4
**Significance:** 3
**Originality:** 3
**Overall Recommendation:** 5
**Confidence:** 4

**Summary:**

LABO (LLM-Accelerated Bayesian Optimization) is a dual-fidelity BO method. LABO models the true objective with a KOH decomposition, a linear combination of a low-fidelity GP surrogate and a discrepancy term. The low-fiedlity GP is fit to LLM predictions, and the full model is fit to experimental observations. A heuristic variance-based "gate" determines when it is necessary to make an expensive experimental measurement. Regret analysis shows that the gate focus observations on a reduced region of the parameter domain where experiments both reduce uncertainty and not amenable to estimatation by the LLM. In several low-dimensional optimization tasks, LABO outperforms vanilla BO and other LLM-incorporating BO methods.

**Compliance With Llm Reviewing Policy:**

Affirmed.

**Final Justification:**

I am satisfied with the rebuttal.

**Key Questions For Authors:**

- Is code available? If not, how could your work be reproduced?
- What LLM(s) did you use? Are your results sensitive to this choice? Should one expect results to improve as LLMs improve?
- Does tau need to be tuned to the optimization problem? To the LLM? To both?

**Limitations:**

Technical limitations of the work are not discussed thoroughly enough. Getting an LLM to produce numerical values within a bounding box might not be trivial as the number of parameter grows. What if there are more complex parameter constraints? What about mixed-type parameters? Larger numbers of observations? Do all of the observations need to fit in context?

**Strengths And Weaknesses:**

The paper is well-organized and clearly written. It creates an algorithmic and theoretical framework for incorporating a modern, important technology (LLMs) into Bayesian optimization.

The paper is light on method, however. It is not clear from the main text how they coax a prediction out of an LLM. The appendix offers guidance on prompting, but given the rapidly-evolving nature of LLMs, the paper would be stronger with (i) a concrete example of a prompt and the output generated, (Ii) some discussion in the main text. Additionally, the prompt seems to end abruptly with "CRITICAL FORMATTING REQUIREMENTS". Is this an error? All that being said, I think an important, but underemphasized, point is that the details of how the LLM produces predictions ultimately don't matter; the LABO model and analysis still apply. This is a significant contribution to the LLM+BO body of research.

The empirical improvements seem modest, and most problems are so low-dimensional as to be uninteresting. I would understand an argument that this not where the value of the paper lies, but no such argument is made.

Also, there is no mention of computer code. Making code available not only aids reproducibility but also aids understanding. There are inevitably myriad small decisions that are made during the production of this kind of work that are not explicit in a paper but, without which, code could not run. This makes code an invaluable resource for understanding and extending your work. For example, how do you define "Vanilla BO"? While the description in the paper is accurate, it is not precise. Did you use one of the many open source libraries? Did you write custom code? How did you sample candidates for or otherwise optimize the acquisition function? What kernel did you use? Etc. The code would contain answers to all these questions.

---

> ### Author Rebuttal · Authors · 2026-03-31
>
> We thank the reviewer for the careful evaluation and constructive suggestions. The comments helped us clarify several important aspects of the paper, which we address below.
>
> **Weakness in method**
>
>
>
> We agree that the current manuscript leaves the LLM prediction process insufficiently specified. In our implementation, the LLM is prompted with the task background, parameter definitions, optimization goal, historical data, and query points, and returns predictions in structured JSON. In the revision, we will add a concise prompt description in the main text and representative examples in the appendix.
>
> “CRITICAL FORMATTING REQUIREMENTS” is not an error. It was used as a generic shorthand for explicit formatting instructions, which were not expanded in the submitted version. We will provide the full task-specific content in the revision.
>
> **Weakness in improvements and task dimensionality**
>
>
>
> Thank you for raising this point. Our target setting is expensive scientific optimization, where the primary goal is to reduce the number of optimization rounds and thus save experimental cost. From this perspective, while the final improvement in the best objective value may appear modest on some tasks, LABO reaches the same target performance in fewer experimental rounds. This reduction in trial-and-error burden is itself practically valuable in scientific discovery. We will clarify this in the revised manuscript.
>
> We added an experiment on the **86-dimensional** superconductor task [1], where the objective is to maximize the superconducting critical temperature. The results below show that LABO remains effective in this higher-dimensional scientific optimization setting. Full results will be included in the revised manuscript.
>
> |   Method   | FINAL OBJ ($\uparrow$) |  ITERS TO 90%   |
> | :--------: | :--------------------: | :-------------: |
> |  **LABO**  |    **91.06 ± 8.05**    | **24.4 ± 23.3** |
> |   LLAMBO   |     72.20 ± 19.78      |   51.2 ± 26.9   |
> |   BOPRO    |      80.38 ± 8.37      |   34.6 ± 31.7   |
> |    CAKE    |     76.57 ± 10.82      |   36.6 ± 26.6   |
> | Vanilla BO |     57.96 ± 18.33      |   64.6 ± 14.3   |
>
> [1] Hamidieh, K. (2018). A data-driven statistical model for predicting the critical temperature of a superconductor. *Computational Materials Science*, *154*, 346-354.
>
> **Weakness in computer code**
>
>
>
> We thank the reviewer for raising this point. The code is already included in the **Supplementary Material**, and in the revision we will make its availability more explicit and provide a clearer implementation summary.
>
> **Question 1**
>
> This point is covered above in **Weakness in computer code**.
>
> **Question 2**
>
> As stated in **Lines 297–299**, we use Intern-S1 in the main experiments because it is pretrained on scientific knowledge and is therefore better aligned with the prediction tasks considered in this work. We already examined the sensitivity of LABO to the LLM backbone in the ablation study by comparing models of different sizes and capabilities. LABO consistently improves performance across different LLMs, while stronger models generally yield larger gains. This suggests that LABO is not tied to a specific model and can further benefit from more capable LLMs. We will clarify these points in the revision.
>
> **Question 3**
>
> In the current paper, $\tau$ is not tuned separately to the optimization problem or the LLM. We use the same fixed $\tau=0.75$ across the main experiments, as stated in **Lines 292–293**. The $\tau$ ablation shows that LABO is reasonably robust to this choice . We therefore keep $\tau$ fixed to avoid extra tuning and to keep the framework simple and reproducible.
>
> **On broader applicability and practical scope**
>
> We thank the reviewer for raising these important points. Bounded numerical prediction is enforced at the prompt level, and our added **86D superconductor task** suggests that LABO remains practical in substantially higher-dimensional settings. More generally, for complex constraints and mixed-type parameters, LABO is compatible with the same feasibility-handling strategies used in BO. Our current experiments already include integer-valued parameters in the **Flow Battery** task. Regarding history length, LABO already works well with limited observations, and a larger history would likely make the LLM-fidelity signal more informative. At the same time, this does not require all observations to be included in context at every step. In practice, only the most relevant subset or a compressed summary can be passed to the LLM, while the GP surrogate retains the full history. We will clarify these points in the revised manuscript.

---

> > ### Author Rebuttal · Reviewer_y8XY · 2026-04-03
> >
> > > Bounded numerical prediction is enforced at the prompt level,
> > [Constraining LLM output](https://github.com/Saibo-creator/Awesome-LLM-Constrained-Decoding) is a well-known, non-trivial problem as is what is called ["hallucination"](https://github.com/LuckyyySTA/Awesome-LLM-hallucination). This reviewer does not believe that "[enforcement] at the prompt level" can have any meaningful interpretation.
> >
> > > In practice, only the most relevant subset or a compressed summary can be passed to the LLM,
> > Without a define protocol, how can it be clear to the reader what is being evaluated?

---

> > > ### Author Response · Authors · 2026-04-05
> > >
> > > Dear Reviewer y8XY,
> > >
> > > We sincerely thank you for your thoughtful follow-up questions. They help us clarify several important aspects of the method and its evaluation. Below, we respond to each point in turn.
> > >
> > > **Question 1**
> > >
> > > We thank the reviewer for the helpful links and will discuss the relevant prior work in the revision.
> > >
> > > We clarify that, in LABO, each LLM query returns a structured response containing the queried candidate features together with a single scalar target prediction. The bounded numerical prediction in our setting concerns the target field, whose value is required to lie within the task-specific valid interval. In implementation, each returned response is checked for both the required JSON format and the valid numeric range before it is used. If the response is malformed or out of range, it is discarded and the LLM is queried again. Only in the very rare case that repeated re-queries still fail, after up to 5 attempts in our implementation, do we replace the invalid response with the posterior mean of the LLM-fidelity GP at that point. Malformed or out-of-range outputs are therefore excluded from the optimization loop. We will make this procedure explicit in the revision.
> > >
> > > Regarding hallucination, LABO does not assume that the LLM prediction is always accurate or reliable. Instead, once a valid scalar prediction is obtained, it is treated as an imperfect LLM-fidelity signal rather than as ground truth. We then use KOH calibration and discrepancy modeling to account for systematic bias and residual error in that signal, while the gating criterion triggers real-fidelity evaluation when the LLM-fidelity signal is unreliable. Recent advances in LLM hallucination mitigation are complementary to LABO and provide a natural direction for future extensions. We will clarify these points more explicitly in the revision.
> > >
> > > **Question 2**
> > >
> > > We thank the reviewer for raising this insightful point. We clarify that, in all experiments reported in this paper, the full accumulated observation history is provided to the LLM. Current LLMs can already handle relatively long inputs well, so providing the full accumulated observation history is feasible in our experiments and gives the LLM more complete information from the optimization process.
> > >
> > > Our earlier remark about using only the most relevant subset or a compressed summary was intended to describe a possible extension for substantially longer histories, rather than the setting evaluated in this paper. In that case, one may introduce a fixed history-selection or summarization protocol to keep the LLM context concise and well defined. We will clarify this more explicitly in the revision. We thank the reviewer again for this helpful comment.
> > >
> > > Best regards,\
> > > The Authors

---

### Official Review · Reviewer_rMvw · 2026-03-04

**Soundness:** 3
**Presentation:** 3
**Significance:** 3
**Originality:** 3
**Overall Recommendation:** 5
**Confidence:** 4

**Summary:**

This paper introduces LABO, a framework that accelerates Bayesian optimization for scientific discovery by treating LLM predictions as a cheap, low-fidelity signal. To integrate this signal, the authors employ the Kennedy-O'Hagan multi-fidelity framework, which models the true objective function as a combination of the LLM's prediction and a learned discrepancy term. The core mechanism is a dynamic gating criterion based on a "discrepancy dominance ratio". For any given candidate, if the uncertainty is primarily driven by the LLM's inherent discrepancy from the truth, the system triggers an expensive real-world evaluation. Conversely, if the uncertainty is simply due to a lack of LLM queries, it relies on the cheap LLM approximation. The pipeline also includes a structured warm-start phase to seed the initial search space with LLM-proposed candidates. The authors provide theoretical guarantees, establishing a cumulative regret bound that decomposes the search into real-fidelity and LLM-fidelity evaluations. Empirically, the method is evaluated across six diverse scientific tasks (ranging from 3 to 20 dimensions) and demonstrates faster convergence compared to vanilla Bayesian optimization and recent LLM-guided baselines under fixed experimental budgets.

**Compliance With Llm Reviewing Policy:**

Affirmed.

**Final Justification:**

I think this paper should be accepted for all of the reasons indicated in my initial review. I initially gave a score of 4, and have now updated my score to 5 to reflect that the author's rebuttal addressed all of my questions and concerns.

**Key Questions For Authors:**

**Q1:** How sensitive is performance to tau? Table 1 sweeps 0.60-0.85 on two tasks, but I'd want to see whether a fixed tau = 0.75 is robust across all six benchmarks or whether task-specific tuning helps. Have you considered adaptive strategies for tau, e.g., adjusting it based on observed LLM-real correlations as optimization progresses?

**Q2:** How well does the KOH linear scaling assumption actually hold for LLM predictions? What happens when the LLM is systematically wrong or negatively correlated with ground truth?

**Q3:** For truly novel chemical spaces with sparse literature coverage, how effective is the LLM warm-start? The method's value proposition depends on the LLM having useful prior knowledge, which may not exist for frontier discovery problems?

**Limitations:**

yes

**Strengths And Weaknesses:**

### Soundness
**Strengths:**
The paper is technically solid. The Kennedy-O'Hagan (KOH) framework is well-established, and its application here is highly appropriate. The gating criterion is intuitive, it measures how much of the predictive uncertainty comes from model-reality mismatch versus LLM uncertainty itself, and it triggers real-world experiments only when the LLM signal is insufficient. The regret analysis in Appendix A is careful, decomposing the regret into three clean terms. I also appreciated that the theory places no structural assumptions on the LLM oracle (allowing it to be inaccurate or inconsistent), which makes the guarantees more broadly applicable.

**Weaknesses:** 1) The linear scaling assumption ($f_R = \rho * f_L + \delta$) may not hold well for LLM predictions in practice, as LLMs can exhibit nonlinear biases that a single scalar $\rho$ cannot correct.
2) The paper assumes LLM inference cost is negligible relative to real experiments. While this is reasonable for physical wet-lab chemistry, it is less obvious for computationally cheap simulations.

### Presentation
**Strengths:**
The writing is clear overall, with a well-designed workflow diagram (Figure 1) and a complete algorithm box. The explicit distinction between "LLM-in-the-loop" methods (which modify BO internals) and "LLM as an external signal" (LABO's approach) provides very helpful framing for the reader.

**Weaknesses:** 1) Prompt engineering is critical to this method's success, yet it is entirely relegated to Appendix B. At a minimum, the main text should discuss the core prompt design choices and their impact.
2) The choice of Intern S1 241B as the default LLM is stated early on but is not justified until the ablation study.

### Significance
**Strengths:**
The paper addresses a real need, as scientific optimization is expensive and data-limited. The gating mechanism is a genuinely useful contribution that goes beyond prior work. For example, [LLAMBO (ICLR 2024)](https://arxiv.org/abs/2402.03921) uses LLMs mainly for initialization and proposals but always evaluates with real fidelity; [BOPRO (ICLR 2025)](https://openreview.net/forum?id=aVfDrl7xDV) updates prompts iteratively but lacks explicit fidelity modeling; and [CAKE (NeurIPS 2025)](https://arxiv.org/abs/2509.17998) injects LLM-derived priors into kernels but does not use LLMs as surrogates. LABO's dynamic fidelity selection fills a clear gap in this landscape.

**Weaknesses:** 1) Looking at the optimization curves in Figure 2, the main advantage is faster convergence rather than dramatically better final objective values.
2) The evaluation spans 3D to 20D problems. I would like to see how this approach scales to genuinely high-dimensional settings (50D+).
3) There is no cost-benefit analysis that explicitly accounts for LLM API/inference costs alongside the experimental costs.

### Originality
The discrepancy dominance ratio as a dynamic gating mechanism is novel and well-motivated. Combining the KOH multi-fidelity framework with LLM predictions is a new idea, as is extending multi-fidelity BO regret theory to this specific setting. While individual components (the KOH framework, LLMs for optimization, multi-fidelity switching) are established in the literature, their synthesis into the LABO pipeline is clearly original (as far as I am aware).

---

> ### Author Rebuttal · Authors · 2026-03-31
>
> We thank the reviewer for the careful reading and constructive suggestions. Below we address the main concerns and will revise the manuscript accordingly.
>
> **Weaknesses in Soundness**
>
> **(1) On the linear scaling assumption**. Our method does not assume that all bias can be corrected by $\rho$ alone. In
> $$
> f_R(x)=\rho f_L(x)+\delta(x),
> $$
> $\rho$ captures global scale alignment, while $\delta(x)$ models the remaining nonlinear bias. We will clarify this more explicitly in the revision.
>
> **(2) On the cost assumption**. LABO’s advantage is not limited to low inference cost. LLMs can draw on broad and cross-disciplinary scientific knowledge to provide useful intuition and directional guidance. By contrast, simulation depends critically on the underlying task-specific model. Even when simulation runtime is inexpensive, building such an accurate model often requires substantial domain expertise, development time, and human effort for each new problem. We will clarify this scope more carefully in the revised manuscript.
>
> **Weaknesses in Presentation**
>
> **(1) On the prompt**.  In the revision, we will summarize the prompt design in the main text and keep the full template in the appendix for reproducibility.
>
> **(2) On Intern S1**. We agree that the motivation for using Intern S1 241B should appear earlier. We use it as the default model because it is pretrained for scientific domains and is therefore better aligned with the tasks studied here. We will explain this choice more clearly in the experimental setup.
>
> **Weaknesses in Significance**
>
>
> (1) **On faster convergence rather than much better values**. Our target setting is expensive scientific optimization, where reducing the number of experimental rounds is itself a primary objective. From this perspective, faster convergence directly lowers the trial-and-error cost of scientific discovery.
>
> (2) **On high-dimensional settings**. We added an **86 dimensional superconductor task** with 80 optimization iterations, on which LABO remains effective. Due to the length limit, we kindly refer the reviewer to our response to **Reviewer y8XY** for the results, which will also be included in the revised manuscript.
>
> (3) **On cost-benefit analysis**. We thank the reviewer for raising this important point regarding the cost-benefit analysis. In the revision, we will compare the total cost required for LABO and vanilla BO to reach the same performance level across all tasks. On the COF task, under a conservative cost setting of 1 USD per LLM query and 100 USD per real experiment, the results are as follows.
>
> |   Method   | Rounds to reach vanilla BO final performance | Number of LLM queries | Total cost/$ |
> | :--------: | :------------------------------------------: | :-------------------: | :----------: |
> | Vanilla BO |                      30                      |           0           |     6000     |
> |    LABO    |                      4                       |          164          |     964      |
>
> **Question 1**
>
>
>
> As stated in **Lines 292–293**, LABO uses a fixed $\tau=0.75$ across all experiments. Our ablation study shows that smaller $\tau$ under-utilizes the LLM signal, while larger $\tau$ over-relies on it.
>
> While adaptive $\tau$ may appear appealing, in our setting it introduces several practical issues. First, estimating reliable LLM–real correlations online is inherently noisy due to the limited number of real-fidelity observations, which can lead to unstable or even misleading updates. Second, adapting $\tau$ per task or during optimization complicates the framework and reduces comparability across benchmarks.
>
> In contrast, a fixed $\tau$ provides a robust and consistent trade-off across all six tasks, leading to stable performance while keeping the method simple and reproducible. We will clarify this design choice in the revised manuscript.
>
> **Question 2**
>
> On the flow battery task, the correlation between LLM-fidelity and real-fidelity is **$R^2=0.649$**
>  with Pearson
> **$r=0.806$**, indicating a fairly strong positive linear association. We will include the corresponding results for all tasks in the revised manuscript.
>
> When the LLM is systematically wrong or negatively correlated with the ground truth, the benefit of LABO decreases and the method relies more on real-fidelity queries. This case is already reflected in our **random-fidelity ablation**.
>
> **Question 3**
>
>  In novel chemical spaces and frontier settings, LABO can still benefit from LLM warm-start and remain effective. Scientific discovery is cumulative rather than occurring in a knowledge vacuum. For example, the theory of relativity was built on the foundations of optics and classical mechanics. Even frontier problems are usually linked to broader scientific knowledge, from which the LLM can still extract useful guidance. In this sense, LABO does not rely on exact prior coverage of the target space, but on broader transferable knowledge. We will clarify this point in the revised manuscript.

---

> > ### Author Rebuttal · Reviewer_rMvw · 2026-04-03
> >
> > The authors addressed each of my concerns and answered each of my questions from my review. I will update my score accordingly.

---

> > > ### Author Response · Authors · 2026-04-04
> > >
> > > Dear Reviewer rMvw,
> > >
> > > Thank you for your thoughtful feedback. We sincerely appreciate your acknowledgement that our rebuttal addressed your concerns and your decision to update your score accordingly. We are grateful for your time and careful evaluation of our work.
> > >
> > > Best regards,\
> > > The Authors

---

### Official Review · Reviewer_n8X2 · 2026-03-06

**Soundness:** 4
**Presentation:** 3
**Significance:** 3
**Originality:** 3
**Overall Recommendation:** 5
**Confidence:** 4

**Summary:**

The paper proposes LLM-Accelerated Bayesian Optimization (LABO), a dual-fidelity Bayesian Optimization framework that integrates Large Language Models (LLMs) into the optimization loop to reduce the number of costly real-world experiments. LABO leverages a Kennedy-O'Hagan (KOH) joint Gaussian Process surrogate to decompose the objective function into a scaled LLM-fidelity prediction and an additive discrepancy term. A dynamic, uncertainty-based gating criterion dictates whether a candidate point should be evaluated via a low-cost LLM query or a high-cost real-world experiment. The authors provide theoretical guarantees demonstrating sub-linear cumulative regret with the potential of improved sample efficiency compared to standard GP-UCB given a sufficiently informative LLM. Extensive empirical evaluations across diverse scientific optimization tasks show that LABO achieves state-of-the-art performance compared to recent baselines under fixed experimental budgets.

**Compliance With Llm Reviewing Policy:**

Affirmed.

**Final Justification:**

This paper introduces a novel framework (LABO) that effectively integrates Large Language Models into Bayesian Optimization using a Kennedy-O'Hagan joint Gaussian Process surrogate. The approach is technically sound and addresses a significant problem in scientific discovery by systematically reducing the need for costly real-world experiments. The theoretical regret bounds establishing the efficiency of the gating mechanism are strong, and the empirical evaluations across diverse scientific tasks are robust.

In my initial review, I raised concerns regarding the use of an OLS approach for estimating the scaling factor instead of a standard joint Marginal Likelihood Estimation, the over-broad dismissal of vanilla BO in high dimensions, and the omission of recent direct LLM-BO literature (e.g., FIBO, ToSFiT).

The authors response fully resolved these concerns. They justified their OLS estimation as a lightweight plug-in that preserves the validity of the covariance structure. They thoroughly addressed the missing literature, providing an accurate theoretical distinction between LABO and FIBO, as well as new, compelling empirical evidence demonstrating LABO's competitiveness/superiority with ToSFiT on a discrete optimization task. They also appropriately acknowledged the complementary nature of recent advancements in vanilla BO priors (Hvarfner et al., 2024) and committed to fixing the minor presentation issues.

Overall, the rebuttal reinforced my prior positive assessment. The methodology is sound, the empirical evidence is convincing, and the authors demonstrated a thorough command of the literature in their response. I confidently recommend this paper for acceptance.

**Key Questions For Authors:**

1. Estimation of $\rho$: Why was a sequential OLS approach chosen to estimate the scaling factor $\rho$ instead of the standard joint Marginal Likelihood Estimation (MLE) typically used for Kennedy-O'Hagan models? Does this impact the rigorousness of the covariance structure?

2. Related Work on Direct LLM BO: How does LABO position itself against recent works that bypass conventional acquisition functions entirely by allowing LLMs to directly sample optimums or perform Thompson sampling (e.g., Sutter et al., 2025 https://arxiv.org/abs/2505.23913; Menet et al., 2025 https://arxiv.org/abs/2510.13328)?

3. Vanilla BO Baseline: Can you address the recent findings (e.g., Hvarfner et al., 2024 https://arxiv.org/abs/2402.02229) that standard BO performs well in high dimensions, and clarify if your vanilla BO baseline  utilized optimized hyperparameter priors to ensure a fair comparison?

**Limitations:**

yes

**Strengths And Weaknesses:**

**Soundness:**

*Strengths:*

The submission is technically very strong. The theoretical results are particularly impressive, establishing solid regret bounds that formalize the efficiency gains of the proposed gating mechanism. Furthermore, the experimental evaluation is robust, demonstrating state-of-the-art performance across six distinct scientific tasks.

*Weaknesses:*

In Section 4.1 and Algorithm 1, the scaling factor ρ is estimated via ordinary least squares (OLS) prior to fitting the discrepancy model. Standard practice for Kennedy-O'Hagan models is to estimate $\rho$ jointly with GP hyperparameters via Marginal Likelihood Estimation. Additionally, the authors claim "data scarcity at the beginning of optimization impedes early-stage exploration". This framing is imprecise; data scarcity actually impedes informed exploration and early-stage exploitation.

**Presentation:**

*Strengths:*

The paper is generally well-structured and the narrative is logical. The methodological workflow is clearly explained.

*Weaknesses:*

There are a few noticeable presentation issues:
- Missing foundational citations: Expected improvement and upper confidence bound (UCB) are introduced in Section 3 without appropriate foundational citations.

- Figure quality: The figures in the experimental section appear to be of low quality. The authors should use vector graphics for final publication.

- Typos: In Section 4.1, "Specialty, We..." should be "Specifically, we...". In Algorithm 1, the inputs incorrectly list the derived composite kernel $k_r$ instead of the discrepancy kernel $k_\delta$.

**Significance:**

*Strengths:*

The paper addresses a highly relevant problem in scientific discovery where real-world evaluations are expensive. Demonstrating practical utility and efficiency gains in domains like chemistry and materials science is a strong contribution.

**Weaknesses:**

The authors overstate the failure of standard methods, claiming that "as the input dimensionality increases, the performance of vanilla BO methods begins to deteriorate significantly". This ignores recent literature (e.g., Hvarfner et al., 2024 https://arxiv.org/abs/2402.02229) showing that properly configured vanilla BO remains highly competitive in high-dimensional spaces.

**Originality:**

*Strengths:*

Integrating LLMs via a KOH multi-fidelity GP with a discrepancy-driven gating criterion is a creative, novel, and well-articulated approach to the problem.

*Weaknesses:*

The Related Work section claims that in existing LLM-in-the-loop BO methods, "final decisions remain governed by conventional acquisition functions". This presents a false dichotomy by ignoring an emerging paradigm where LLMs directly generate acquisition maximizers or perform Thompson sampling (e.g., Sutter et al., 2025 https://arxiv.org/abs/2505.23913; Menet et al., 2025 https://arxiv.org/abs/2510.13328).

---

> ### Author Rebuttal · Authors · 2026-03-31
>
> We thank the reviewer for the careful reading and thoughtful comments, as well as the positive assessment of the technical quality, theoretical analysis, and empirical evaluation. Below we address the main concerns and will revise the manuscript accordingly.
>
> **Weaknesses in Soundness and Question 1**
>
>
>
> **(1) Estimation of $\rho$.** We agree that, in standard Kennedy-O’Hagan modeling, $\rho$ is typically estimated jointly with the GP hyperparameters via marginal likelihood. In our implementation, $\rho$ is instead estimated by OLS for practical reasons. In LABO, the surrogate is repeatedly refit as new observations arrive, and $\rho$ serves mainly as a scalar alignment coefficient between LLM-fidelity and real-fidelity observations. OLS therefore provides a lightweight plug-in estimate that is easy to recompute during iterative updates.
>
> This choice does not affect the covariance construction. Once $\rho$ is fixed, the surrogate still takes the KOH form:
> $$
> f_H(x)=\rho f_L(x)+\delta(x),
> $$
> which induces the usual cross-covariance and real-fidelity covariance
> $$
> \mathrm{Cov}(f_H(x),f_L(x'))=\rho\,k_L(x,x'), \qquad
> \mathrm{Cov}(f_H(x),f_H(x'))=\rho^2 k_L(x,x')+k_\delta(x,x').
> $$
> Accordingly, for any fixed scalar $\rho$, the induced covariance remains valid when $k_L$ and $k_\delta$ are valid kernels. We will clarify this implementation choice and the standard joint MLE more explicitly in the revision.
>
> **(2) Data scarcity statement.** We thank the reviewer for pointing this out and will revise the phrasing accordingly.
>
> **Weaknesses in Presentation**
>
>
>
> We thank the reviewer for pointing out these presentation issues. We will address them in the revision by adding the appropriate foundational citations, replacing the current figures with higher-quality vector graphics, correcting the typo in Section 4.1, and fixing the kernel notation in Algorithm 1.
>
> **Weaknesses in Significance and Question 3**
>
>
>
> We thank the reviewer for pointing us to Hvarfner et al. (2024). We view LABO as complementary to this line of work. LABO adds an LLM-fidelity signal and a gating mechanism, but still relies on GP surrogate modeling over real-fidelity observations and residuals. Improved GP priors and calibration strategies from Hvarfner et al. (2024) can therefore, in principle, be incorporated into both vanilla BO and LABO. In our experiments, LABO and vanilla BO share the same GP modeling pipeline, which keeps the comparison fair. We will revise the manuscript accordingly and discuss this paper explicitly in the main text.
>
> **Weaknesses in Originality and Question 2**
>
>
>
> We thank the reviewer for highlighting these recent works.
>
> Sutter et al. propose FIBO, which leverages a pretrained generative model to accelerate acquisition optimization and sampling within BO, thereby reducing the computational overhead per iteration while maintaining similar optimization performance. In contrast, LABO targets a different regime, namely scientific optimization problems where the dominant cost arises from expensive real-world evaluations rather than acquisition optimization. Accordingly, LABO is designed to reduce the number of expensive real-world evaluations, whereas FIBO focuses on reducing the computational cost per iteration. We will clarify this distinction in the related work to better position LABO with respect to FIBO.
>
> Menet et al. propose ToSFiT for discrete optimization, where Thompson sampling is reformulated as an online LLM fine-tuning process. We compared LABO with ToSFiT on a discrete optimization task involving CBD lipid nanoparticle formulations [1]. On this task, LABO achieves better final performance and faster convergence than both ToSFiT and vanilla BO. We will include these comparisons in the revision.
>
> |   Method   | FINAL OBJ ($\uparrow$) | ITERS TO 90%  |
> | :--------: | :--------------------: | :-----------: |
> |  **LABO**  |   **0.898 ± 0.017**    | **4.4 ± 3.4** |
> |   ToSFiT   |     0.883 ± 0.017      |   6.0 ± 5.5   |
> | Vanilla BO |     0.711 ± 0.016      |   6.4 ± 2.8   |
>
> [1] Xie, Y, et al. (2024). CBD-loaded nanostructured lipid carriers: Optimization, characterization, and stability. *ACS omega*, *9*(39), 40632-40643.

---

> > ### Author Rebuttal · Reviewer_n8X2 · 2026-04-01
> >
> > The authors have satisfactorily addressed my concerns and reaffirmed my decision to recommend acceptance.

---

> > > ### Author Response · Authors · 2026-04-02
> > >
> > > Dear Reviewer n8X2,
> > >
> > > Thank you for your constructive feedback throughout the review process and for acknowledging that our responses have addressed your concerns.
> > > We are delighted that our clarification has reaffirmed your decision to recommend acceptance. We sincerely appreciate your time and thoughtful evaluation of our work.
> > >
> > > Best regards, \
> > > The Authors

---

### Official Review · Reviewer_KYoU · 2026-03-10

**Soundness:** 3
**Presentation:** 2
**Significance:** 2
**Originality:** 3
**Overall Recommendation:** 4
**Confidence:** 4

**Summary:**

The paper proposes LABO, an LLM-accelerated Bayesian Optimization (BO) method. The key idea is to use an LLM to make predictions for the objective function values and then construct a GP surrogate model from the LLM-generated dataset. The paper also proposes a mechanism to construct another GP surrogate model to calibrate for the discrepancy between the LLM predictions and the ground-truth function values, as well as a gating mechanism to select when to use LLM predictions. Theory is also developed for the proposed method. Experiments are conducted on various optimization tasks in the applied science domain.

**Compliance With Llm Reviewing Policy:**

Affirmed.

**Final Justification:**

As mentioned in my rebuttal acknowledgement, most of my concerns are addressed, especially the novelty/contribution of the work, as well as the theoretical analysis (although I could not go into very detail, but the answers seem to be overall reasonable). For this, I will increase my score to 4 - the reason I won't be able to increase higher is that (1) I still don't feel the idea to be very exciting, and (2) the existing experiments are not very satisfying (many optimization tasks are not the benchmark ones, and the budgets are too small).

**Key Questions For Authors:**

1.	From Lines 259-262, it says that LABO selects a batch of candidate points by maximizing an acquisition function, but this could be inefficient because these data points could be very close to each other. In the literature, people use some techniques like Kriging Believer, etc to make sure the batch candidate points are not very close to each other.
2.	In Section 5, the paper states that “we prove that this partition stabilizes after a finite number of steps, giving rise to a limiting real-fidelity region X*_R”, but I don’t see any theorem associated with this in the paper. Is it part of Theorem 5.1?
3.	Theorem 5.1 relies on T*_R as the number of real fidelity queries within the limiting region X*_R, but there has been no proof regarding the existence of X*_R. Furthermore, if this happens, this seems to only occur when the iteration T reaches a very large value, so the bound provided in Theorem 5.1 is only applicable for large T? If yes, then how large T can be? If no, then can the authors answer my concern mentioned? Furthermore, the bound in Theorem 5.1 does not provide the upper bound on the maximum mutual information \Psi_T, so we don’t know if the cumulative regret R_T is sublinear.
4.	I’m just wondering if there are any guidelines on how to choose the gating threshold \tau in practice?

**Limitations:**

The paper doesn’t mention much about the limitations of the proposed approach.

**Strengths And Weaknesses:**

Strengths:
+ The paper proposes an LLM-guided BO approach, which is surely a timely topic, given the rise of LLMs' capabilities nowadays.
+ The paper seems to be the first work that directly uses an LLM to predict the objective function evaluation and construct a dataset from this to fit a GP surrogate model.
+ The writing is generally easy to read.

Weaknesses:
+ The proposed method does not seem very exciting to me; it only makes use of the LLM to generate the predictions for the objective function evaluation, and a gating threshold to decide when to trust the LLM prediction.
+ I’m not sure if the developed theory is correct; there are some terms that are not yet proved, e.g. X^*_R. Also, there is no bound on the maximum mutual information \Psi_T(), so there is no guarantee that the developed bound is sub-linear. Please see my questions for more detailed information.
+ For the experiments, first, the optimization tasks used in the paper are from many applied science domains, which is good, but none of these is even from a machine learning task (e.g. AutoML). Also, these optimization tasks are not the common benchmarks used to evaluate BO algorithms. Furthermore, the number of iterations is very small, only 30, which is very limited; normally, we evaluate at least 10d (d is the optimization problem dimension).
+ In the experiments, the gating threshold \tau is set to 0.75 and based on the results in Table 1, it seems this is the value that yields the best performance of the proposed method. For other choices, the performance of the proposed method seems to be lower, and these results might not really be much better than existing LLM-based BO baselines. In practice, it’s normally the case that we can’t find an optimal value for \tau so this puts a lot of constraints on the performance of the proposed method.

---

> ### Author Rebuttal · Authors · 2026-03-31
>
> We thank the reviewer for the careful reading and constructive comments. Below we address the main weaknesses and then the remaining key questions.
>
>
>
> **Weakness 1**
>
>
>
> We address this point from two aspects.
>
> **(1) Methodological novelty.** The key novelty of LABO is its fidelity-aware gating mechanism. LABO treats LLM outputs as low-fidelity signals and uses a gating mechanism to decide when they can substitute for real-fidelity evaluations. To our knowledge, this calibrated fidelity-aware integration is new in BO.
>
> **(2) Broader significance.** More broadly, LABO takes a step toward connecting knowledge-driven and data-driven optimization in a safe and controlled way. In many scientific domains, useful prior knowledge is unstructured and often imperfect, making it difficult to incorporate into BO. LABO provides a principled interface for bringing such knowledge into the optimization process.
>
>
>
> **Weakness 2**
>
>
>
> We apologize that the main text does not explicitly define $\mathcal{X}_R^*$, which makes the theory hard to follow. The points below also address **Questions 2 and 3**.
>
> **(1) Definition of $\mathcal{X}_R^\*$.** In the appendix, **Definition A.6** (**Lines 715–735**) defines
>
> $$
> \mathcal{X}_R^* = \underset{t \to \infty}{\liminf} \mathcal{X}_R^{(t)}
> $$
>
> **Lemma A.7** shows that there exists a finite $T_0<\infty$ such that, for all $t\ge T_0$, the gating decision at each $x$ stabilizes and matches its membership in $\mathcal{X}_R^\*$. Thus, $\mathcal{X}_R^\*$ is well defined as the stabilized real-fidelity region.
>
> **(2) Relation to Theorem 5.1.** In Theorem 5.1, $T_R^\*$ denotes the number of real-fidelity queries in $\mathcal{X}_R^\*$. Once the finite-step stabilization result of Lemma A.7 is made explicit, the regret decomposition in Theorem 5.1 is well defined.
>
> **(3) Relation between finite-step stabilization and $T$.** We clarify that our theoretical result should be understood as an **asymptotic cumulative regret guarantee**. It is not intended to provide a uniform statement for every finite $T>1$. Instead, it establishes that, under the stated assumptions, LABO achieves **sublinear cumulative regret**, and that LLM guidance can continue to improve the Bayesian optimization process asymptotically.
>
> **(4) $\Psi_T$ and sublinear regret.** As stated in **Remark 5.2**, $\Psi_T$ is the information-gain term in standard GP-UCB analysis. Since $\mathcal{X}_R^\* \subseteq \mathcal{X}$,
> $$
> \Psi_T(\mathcal{X}_R^*) \le \Psi_T(\mathcal{X}).
> $$
> Therefore, once Theorem 5.1 is instantiated with the standard kernel-specific upper bounds on information gain for the induced kernel, it yields sublinear cumulative regret under the usual GP-UCB assumptions.
>
> We will revise the manuscript accordingly and would appreciate the reviewer’s further review.
>
>
>
> **Weakness 3**
>
>
>
> We added two additional **AutoML** tasks, **MLP (5D)** and **SVM (2D)**, from HPOBench [1], both evaluated for **80 iterations**. LABO achieves the best final objective and the fastest convergence on both. We also added an additional **86D** high-dimensional task. Due to the length limit, we kindly refer the reviewer to our response to **Reviewer y8XY** for the corresponding results. Full details will be included in the revised manuscript.
>
> |   Method   | FINAL OBJ $\uparrow$ (SVM) | ITERS TO 90% (SVM) | FINAL OBJ $\uparrow$ (MLP) | ITERS TO 90% (MLP) |
> | :--------: | :------------------------: | :----------------: | :------------------------: | :----------------: |
> |    LABO    |       0.909 ± 0.016        |    33.8 ± 18.8     |       0.941 ± 0.009        |    30.0 ± 10.2     |
> |   LLAMBO   |       0.892 ± 0.023        |    42.0 ± 14.5     |       0.925 ± 0.006        |    50.2 ± 20.8     |
> |   BOPRO    |       0.898 ± 0.019        |     60.2 ± 9.6     |       0.934 ± 0.013        |    47.2 ± 24.9     |
> |    CAKE    |       0.904 ± 0.014        |    50.0 ± 24.5     |       0.939 ± 0.016        |    52.2 ± 15.0     |
> | Vanilla BO |       0.880 ± 0.018        |    42.8 ± 20.1     |       0.905 ± 0.014        |    54.4 ± 16.8     |
>
> [1] Eggensperger, K, et al. HPOBench: A collection of reproducible multi-fidelity benchmark problems for HPO.
>
>
>
> **Weakness 4**
>
>
>
> As stated in **Lines 292–293**, LABO uses $\tau = 0.75$ in all experiments. We will make this more explicit in the revised manuscript. The ablation shows that smaller $\tau$ underuses the LLM signal, while larger $\tau$ over-relies on it, so we recommend $0.75$ as a practical choice. This also addresses **Question 4**.
>
>
>
> **Question 1**
>
> LABO uses sequential greedy batch selection with fantasy updates, rather than repeated maximization on a fixed surrogate. This is similar to Kriging Believer and helps avoid overly clustered batch points. We will clarify this in Lines 259–262.

---

> > ### Author Rebuttal · Reviewer_KYoU · 2026-04-03
> >
> > I'd like to thank the authors for their effort in addressing my concerns. Most of my concerns are addressed, especially the novelty/contribution of the work, as well as the theoretical analysis (although I could not go into very detail, but the answers seem to be overall reasonable). For this, I will increase my score to 4 - the reason I won't be able to increase higher is that (1) I still don't feel the idea to be very exciting, and (2) the existing experiments are not very satisfying (many optimization tasks are not the benchmark ones, and the budgets are too small).

---

> > > ### Author Response · Authors · 2026-04-03
> > >
> > > Dear Reviewer KYoU,
> > >
> > > Thank you very much for your thoughtful feedback. We sincerely appreciate your acknowledgement that our rebuttal addressed most of your concerns, especially regarding novelty and the theoretical analysis, and we are delighted that you increased your score after reading our response. We also value your comments on the remaining points regarding the overall excitement of the idea and the experimental scope.
> > >
> > > Best regards,\
> > > The Authors

---

### Decision · Program_Chairs · 2026-04-30

**Decision:**

Accept (regular)

**Comment:**

This paper considers the problem of how to integrate LLM predictions into Bayesian optimization to accelerate the optimization process. The key idea is to treat LLM predictions as a low-fidelity evaluation and integrate them into a multi-fidelity formulation by modeling the true objective function as a combination of the LLM's prediction and a learned discrepancy term; and use a dynamic gating approach to select candidates for real experimental evaluation. The paper also provides theoretical guarantees and good empirical results.

All reviewers' liked the overall approach and execution. Authors' addressed the questions and comments from reviewers in the rebuttal phase.

I recommend accepting the paper and encourage the authors' to incorporate all the review comments / discussion to improve the final paper to make it useful for the community.